# The ATMONSYS water vapor DIAL: Advanced measurements of short-term variability in the planetary boundary layer

Johannes Speidel[1], Hannes Vogelmann[1], Andreas Behrendt[2], Diego Lange[2], Matthias Mauder[1,3], Jens Reichardt[4], and Kevin Wolz[1,3]

[1]Karlsruhe Institute of Technology (KIT), Institute of Meteorology and Climate Research Atmospheric Environmental Research (IMK-IFU), Campus Alpin, Garmisch-Partenkirchen, 82467, Germany
[2]Institute of Physics and Meteorology (IPM), University of Hohenheim, Stuttgart, 70599, Germany
[3]Dresden University of Technology (TUD), Faculty of Environmental Sciences, Institute of Hydrology and Meteorology, Tharandt, 01737, Germany
[4]Richard-Aßmann-Observatorium, Deutscher Wetterdienst, Lindenberg, 15848, Germany

**Correspondence:** Johannes Speidel (johannes.speidel@kit.edu)

**Abstract.** High-resolution measurements of water vapor concentrations and their transport throughout the turbulent planetary boundary layer (PBL) and beyond are key for an enhanced understanding of atmospheric processes. This study presents data from the mobile atmospheric monitoring system (ATMONSYS) differential absorption lidar (DIAL), operated with a novel Titanium-Sapphire (Ti:Sa) laser concept, for the first time. The ATMONSYS DIAL aims to resolve turbulence throughout the PBL with a sampling frequency of $10\,\mathrm{s}$ and vertical resolutions of less than $200\,\mathrm{m}$. General measuring capabilities during high-noon, clear-sky, summer conditions with a maximum vertical measurement range of $> 3\,\mathrm{km}$ and statistical uncertainties of $< 5\,\%$ are demonstrated. The analysis of turbulence spectra shows good agreement with Kolmogorov's law, demonstrating the system's capability to resolve turbulence. However, deviations from Kolmogorov behavior are observed at certain frequency ranges. By combining the ATMONSYS DIAL with an adjacent high-quality Doppler wind lidar, some of these deviations are mitigated in the co-spectra due to independent noise from both instruments. However, intermediate deviations from Kolmogorov behavior persist, likely due to surrounding surface heterogeneities. The agreement of the co-spectra with Kolmogorov's law at the highest frequencies demonstrates that the ATMONSYS DIAL is capable to resolve turbulent latent energy fluxes down to the measurement's Nyquist frequency of $5 \cdot 10^{-2}\,\mathrm{Hz}$. A system cross-intercomparison of the ATMONSYS DIAL with two adjacent water vapor Raman lidars and radiosondes shows overall good agreement between the sensors, despite minor DIAL deficiencies under certain conditions with broken clouds passing over the lidar. The observed profile-to-profile DIAL fluctuations and sensor-to-sensor deviations, in combination with low statistical uncertainty, highlight the advantage of humidity lidars, such as the ATMONSYS DIAL, to capture both short-term and small-scale dynamics of the lowermost atmosphere.

## 1 Introduction

Accurate and precise water vapor measurements reaching throughout the entire planetary boundary layer (PBL) and into the lower free troposphere are crucially important for an improved understanding of several atmospheric processes. On a large

scale, knowledge about the water vapor distribution is highly relevant for the investigation of climate effects and, therefore, climate modeling. This climatic effect is caused by both the opacity of water vapor in the infrared spectrum and the substantial role of humidity in cloud formation (e.g. Held and Soden, 2000; Schneider et al., 2010; Sherwood et al., 2010; Intergovernmental Panel On Climate Change, 2021). Furthermore, as a secondary effect, cloud formation naturally controls the initiation and distribution of precipitation. Therefore, numerical weather prediction models largely depend on humidity information at high tempo-spatial resolutions in order to improve their precipitation forecast. The improved prediction skills are dependent on better parameterizations of the underlying humidity transport processes (e.g. Behrendt et al., 2011; Santanello et al., 2011; Wulfmeyer et al., 2015, and references therein). Finally, the prevailing water vapor concentration can be understood as an amount of available atmospheric energy, which is stored in form of latent heat. The transport of latent heat can be quantified by the calculation of latent heat fluxes - if accompanying measurements of the wind velocity are available (e.g. Stull, 1988; Foken, 2017). Those fluxes are not only relevant for cloud formation processes but also for the widely discussed problem of the energy balance closure problem (e.g. Mauder et al., 2020b, and references therein).

From the above-mentioned processes, humidity measurements throughout the entire PBL are especially needed for better model parameterizations as well as the calculation of latent energy fluxes. Such measurements have to be taken at high spatio-temporal resolutions. This is largely due to the fact that surface heterogeneity leads to a very fragmented pattern of evapotranspiration and that the transport throughout the PBL is turbulence-driven by eddies with diameters reaching from roughly $10^{-1}$ m to $10^3$ m and not at all dominated by prevailing wind patterns on a synoptical scale (e.g. Wulfmeyer et al., 2018; Santanello et al., 2011; Mauder et al., 2020b). This results in fast-changing humidity distributions not only in the horizontal but also in the vertical scale. Regarding turbulent fluxes, the temporal resolutions have to be even higher, reaching dimensions of at least $\Delta t \leq 10$ s. As already mentioned, flux calculation is only possible if wind observations of the very same air parcels are available. The resolutions for those measurements should, of course, be equally high. Such measurements of vertical fluxes of latent heat throughout the entire PBL are very ambitious and, at the same time, often requested by the modeling community (e.g. Helbig et al., 2021). Due to the mentioned surface heterogeneity, these measurements are preferably undertaken by mobile systems - allowing for measurements at different locations with different surface conditions.

In principle, the vertical fluxes of latent heat could be measured by various systems using different physical approaches. In-situ measurements of latent heat fluxes, using the combination of hygrometers and sonic anemometers have been proven to be possible on high frequencies at high accuracy (e.g. Mauder et al., 2007, 2020a). However, at a fixed geolocation such measurements are limited by the height of possible towers ($\propto 10^2$ m) on which the sensors can be mounted on (e.g. Davis et al., 2003). Deployments of the sensors to radiosondes, balloons or aircraft overcome the issue of insufficient height, but, on the other hand, hinder simultaneous measurements throughout the PBL at the very same location. Passive remote sensing systems like e.g. microwave radiometers can partly overcome these problems but are, however, only available with lower tempo-spatial resolutions that are not capable of measuring turbulent fluctuations.

Lidar measurements, in contrast, as active remote sensing instruments, have the advantage that data with high spatio-temporal resolution can be taken continuously at a fixed location up to high altitudes. This advantage makes them the preferable choice for measurements throughout the PBL - given that the data quality is good enough. Up to this point, to our knowledge, there

have only been very few successful attempts on measurements of the vertical latent heat flux throughout the PBL with a temporal resolution of down to 10s (Behrendt et al., 2020). This is due to the fact that accurate measurements at such temporal resolutions, especially of humidity, are still technologically challenging.

In recent years, there have been several large-scale measurement campaigns, using a high density of different, complementing measurement instruments with the overall goal of an improved, comprehensive understanding of the atmospheric processes inside the PBL and above, as well as the atmospheric interaction with the (heterogeneous) earth surface (e.g. Wulfmeyer et al., 2011; Butterworth et al., 2021; Hohenegger et al., 2023).

The herein presented, in-house developed, mobile Atmospheric Monitoring System Differential Absorption Lidar (ATMON-
SYS DIAL) has been especially designed to enable measurements of water vapor throughout the entire PBL and beyond. The DIAL is aiming at high spatio-temporal resolutions in order to push the current limitations of water vapor measurements throughout the PBL by capturing a large portion of the turbulent transport scales. Within this manuscript we show data from the ATMONSYS water vapor DIAL that has been collected during the Field Experiment on submesoscale spatio-temporal variability in Lindenberg (FESSTVaL) campaign in 2021 (Hohenegger et al., 2023). Based on these data we (1) demonstrate
the general stability of the system over time, its range, and its vertical and temporal resolution during daylight. Also, in order to evaluate the system's suitability for the analysis of turbulent transport, we (2) analyze (co-)spectra of the ATMONSYS DIAL and an adjacent, vertical-staring, Doppler wind lidar. Finally (3), we compare three different high-power humidity lidars in close proximity to each other for the FESSTVaL campaign. In addition, radiosonde ascents from the German Meteorological Service (DWD) are used as a direct reference. Up to now, instrument inter-comparisons have mainly been shown for lidar-lidar
or lidar-radiosonde comparisons (e.g. Behrendt et al., 2007a, b; Bhawar et al., 2011). The possibility of a cross-intercomparison between three high-power humidity lidars including radiosondes directly next to the setup location is a very unique advantage and gives more detailed insight into the capabilities of lidar humidity measurements.

## 2 The ATMONSYS DIAL - instrument description

### 2.1 General design

The ATMONSYS lidar is designed as an experimental mobile system for observations of water vapor, aerosol, and temperature profiles throughout the PBL and beyond. The system has been developed with the goal to observe these measures throughout the entire PBL at such high quality to resolve turbulent changes in their concentrations and values, especially regarding humidity. Therefore, the water vapor DIAL can be seen as the centerpiece of the ATMONSYS system, which is accompanied by an elastic aerosol backscatter lidar and a temperature rotational Raman lidar. The DIAL technique is advantageous for measuring water
vapor for several reasons, most important because it is inherently self-calibrating by its working principle (e.g. Weitkamp, 2005). Measurements at two different but nearby wavelengths are performed quasi at the same time with the same optical geometry. The only difference of these two measurements is practically the differential extinction by water vapor along the light path, while all other instrument dependent parameters cancel out (e.g. Browell et al., 1979; Zuev et al., 1983; Ehret et al., 1996; Wulfmeyer and Walther, 2001). Additionally, the lidar return of the DIAL is backscatter mainly from air molecules

(Rayleigh scattering) and aerosols, which is much stronger than the weak inelastic backscatter used in Raman lidars. This allows for short integration times (<10s) and a full daytime capability over the entire altitude range. A major challenge is the very sophisticated single-mode laser technique which is mandatory for narrowband water vapor DIALs (Wulfmeyer and Bösenberg, 1998; Wulfmeyer, 1998). All three lidars are housed within the same serial 20" cargo container and therefore easily movable by regular cargo trucks (Fig. 1a). Major modifications to the container have been made regarding the ceiling and the

posts that standard cargo container normally stand on (Fig. 1b). In addition to the fixed serial posts, 4 levelling jacks on all corners allow for mechanical adjustments of the container's height and accurate level by hand - independent of flat ground. Two motor-driven flaps in the ceiling allow the system's periscope to be lifted outside the container.

Stable conditions within the container are vital for proper operation of the laser setup. Ideally, this would mean that there are very low temperature changes within the interior and very low vibrations whatsoever. Therefore, the container's room

temperature is controlled by a powerful 10kW air conditioning/heating system which is placed outside the container during operation to prevent disturbing vibrations. All further components inside the container that cause minor vibration due to fan propellers are mechanically decoupled with slings or rubber stands. Therefore, consequently and very importantly, the optical bench inside the ATMONSYS system is also mechanically decoupled by pneumatic leveling on all 4 posts of the optical bench. As the ATMONSYS system is not eye-safe, a safety radar is mounted on the container top with an automated interlock for the

laser in case of any fly-overs. In addition, sensors for wind and rain are mounted next to the safety radar and are also connected with the laser's interlock in case of unexpected weather changes. A battery-powered uninterruptible power supply guarantees for proper shutdown of the system and closing of the ceiling flaps in case of power outage. Also, in the case of a blackout, a block heater prevents the system against freezing temperatures inside the container.

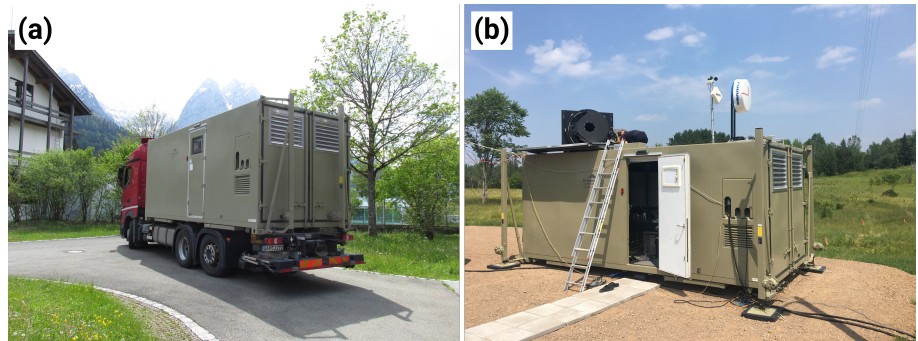

**Figure 1.** (a): The ATMONSYS lidar during truck transport. The swing-out leveling jacks can be seen at the rear side of the container. (b): Set up container with elevated periscope. Also seen is the vertically pointing safety radar as well as the basic meteorological measuring rod, ensuring closing of the ceiling flaps in case of rainfall.

## 2.2 Optical concept

As mentioned above, the ATMONSYS lidar consists of three different lidars for the measurements of water vapor, aerosol, and temperature. A sketch of the general optical concept of both the ATMONSYS lidar emitter and receiver is presented in Fig. 2.

A seeded, diode-pumped, Nd:YAG laser (InnoLas) with a repetition rate of 100 Hz and a maximum average power of $P = 45\,\text{W}$ at the wavelength $\lambda = 1064\,\text{nm}$ is used as the main power source for all three lidar laser emissions. The elastic backscatter aerosol channel and the temperature rotational Raman channel can directly be operated with the emission after the second-/third-harmonic generator (532 nm / 355 nm) respectively. The water vapor measurements are performed with the ATMONSYS DIAL which is operated at two wavelengths in the 817 nm band of water vapor in an alternating sequence. The laser transmission is driven by an in-house developed titanium sapphire (Ti:Sa) laser, which is pumped by a frequency-doubled Nd:YAG laser with $\approx 25\,\text{W}$ at $\lambda = 532\,\text{nm}$. Details on the newly developed transversal pumping configuration and the resonator setup of the Ti:Sa laser as well as its implied advantages are given in Vogelmann et al. (2022). The exact wavelengths of the Ti:Sa are defined by two tunable external cavitiy-diode lasers (DL1/DL2, Fig. 2) which are operated in an injection-seeding procedure with shot-to-shot alternation between $\lambda_\text{on}$ and $\lambda_\text{on}$ similar to Ertel (2004). Whereas the main focus of the ATMONSYS lidar is set towards high power in the water vapor DIAL channel, the temperature and aerosol channels can be considered as a pleasing side product. Therefore, the second-harmonic generator is tuned for maximum power and completely used as the pump light for the Ti:Sa laser. The third-harmonic generator is also tuned to maximum power @355 nm but only takes the non-converted light @1064 nm after the second-harmonic generator. The aerosol channel @532 nm is then operated only with the unconverted shares from the third-harmonic generator as well. In the end, all pathways of the three different wavelengths (817 nm, 532 nm, 355 nm, all s-polarized) are combined congruently before they leave the ATMONSYS container. The average powers of all lasers can be seen in Tab. 1. The option of implementing a beam expander has been omitted due to concerns with changes in polarization as well as chromatic aberrations caused by the three different wavelengths. A possible workaround to this problem would have been the implementation of multiple beam expanders which, in turn, requires a lot of space and bears the downside of added complexity in their proper adjustment.

For lidar operation, a periscope consisting of two slightly oval receiving mirrors ($d_{min} = 64\,\text{cm}$), angled at 45°, is lifted over the container's top. The outgoing laser beams are transmitted by small mirror inlets in the center of the two big mirrors. The periscope has been constructed with the intention of scanning measurement patterns. However, inertial forces related to the high weight of the periscope lead to slight mechanical distortions during movement. Therefore, now, the system is only operated in vertical stare mode. On the receiving end, the backscattered light is collected by two identical Newtonian telescopes ($d = 200\,\text{mm}$, $f = 800\,\text{mm}$). The initial idea of the receiver's design with two telescopes was to establish a near-field channel as well as an identical far-field channel. However, for its use as a boundary layer lidar, one channel is sufficient if the deepest hundred meters above ground are accepted as being blind. It was estimated that an additional near field receiver would lower the start of the lidar range only by roughly 50 m at the most. Instead, the very weak Raman signal return is now taken solely from one telescope in an optically isolated area of the polychromator (Fig. 2 and abbreviations therein). Taking into the account that Raman scattering by $N_2$ and $O_2$ molecules leads to a depolarization ratio of approximately 3/4 (e.g. Penney et al., 1973), a thin film polarizer with high reflection for 355 nm p-polarized light is implemented (S3, Fig. 2b). By this, the polychromator's efficiency for temperature measurements could be improved. The other telescope solely collects the signals for aerosol and water vapor measurements. A dichroic mirror with high transmittance for $\lambda = 600\text{-}850\,\text{nm}$ and high reflectivity at $\lambda = 532\,\text{nm}$ (LASEROPTIK) is used to split up the signal for aerosol and humidity measurements. For the aerosol channel, a 2 inch filter

with the center wavelength of 532.23 nm and a bandwidth of 0.25 nm is used. For water vapor measurements, several filters are available, depending on the chosen absorption line. In the herein presented data we used a filter with 0.5 nm bandwidth and $\lambda_0 = 817.223$ nm. For completeness, we point out that the Raman section of the polychromator in Fig. 2(b) shows a second water vapor detector as well. This channel uses a wider filter width a bandwidth of 2 nm. It has been installed for switching wavelengths without the need of mechanical filter changes. In addition, this allows for using a wider spread of $\lambda_{on}$ and $\lambda_{off}$. Also, by this, potential problems with the angle of incidence in the near field could be investigated and reduced (Bailén et al., 2019). At the focal point of both telescopes, a slit diaphragm (A in Fig. 2) defines the actual field of view and allows for the reduction of background light. The field of view ranges from 2.5 - 5 mrad, depending on the axis of the slit diaphragm. After the telescope's focal point, the light beam is collimated between pairs of $f = 100$ mm plano-convex lenses (L1). After passing its respective interference filters and beam splitters within this collimated path, each beam diameter is again reduced (L1) and collimated by $f = 15$ mm plano-convex lenses (L2) directly in front of photomultiplier tubes (PMT, manufactured by HAMAMATSU).

**Table 1.** Specifications of the laser systems.

| Parameter | Value |
| --- | --- |
| Averaged output power @355 nm | 1.8 W |
| Averaged pump power @532 nm (pumping) | 26 W |
| Averaged output power @532 nm | 1.6 W |
| Averaged output power @817 nm | 2 W |
| Pulse duration @355/@532 nm | 8 ns |
| Pulse duration @817 nm | 50 ns |
| Averaged power of seeding diode lasers | 50 mW |
| Repetition rate | 100 Hz |

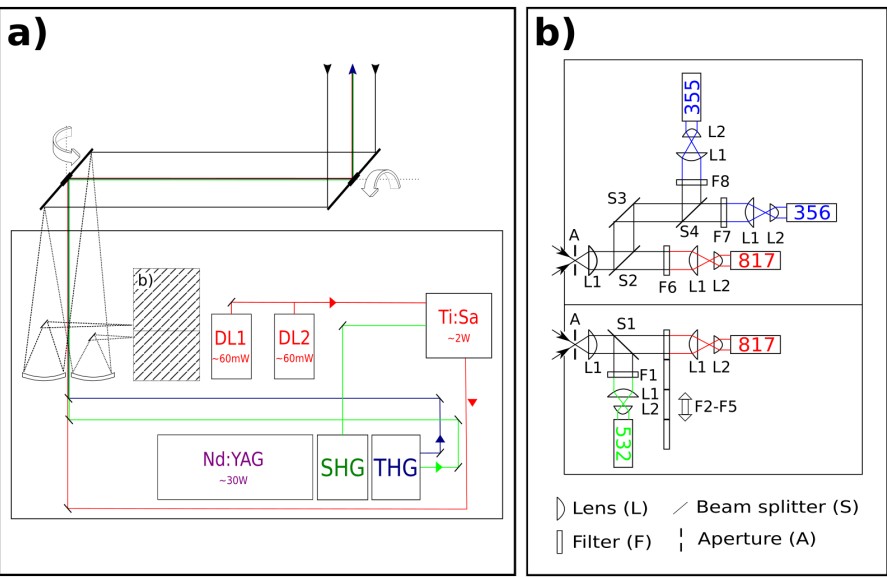

**Figure 2.** Sketch of the overall optical setup. a): Top-view of the optical bench inside the container box with seeding diode lasers (DL), one Nd:YAG laser with second-harmonic and third-harmonic generators (SHG/THG) as well as the Ti:Sa laser system. The outgoing laser emission is transmitted via small mirror inlets in the rotatable periscope mirrors on the roof. The big mirrors ($d_{\min} = 64\,\mathrm{cm}$) direct the back-scattered light towards two receiving telescopes. b): Receiver with light paths for both receiving telescopes.

## 2.3 Signal processing, data retrieval, and statistical uncertainty

The signal from each of the channel's PMTs in the receiver (Fig. 2b) is collected by a 12 bit transient digitizer (Licel) with a memory depth of 24 bit, operated at a voltage range of $500\,\mathrm{mV}$. The $24^{th}$ bit acts as clip flag in the case of signal overflow. However, even if the signal level from a $10\,\mathrm{s}$ integration doesn't indicate overflow, partial overflow due to atmospheric fluctuations can still lead to non-linear signal behavior of the $10\,\mathrm{s}$-integrated signal, even if the clip-bit does not indicate overflow. As a rule of thumb, this effect can already occur if the signal level is higher than $50\,\%$ of the voltage range (private conversation with

B. Mielke, Licel). The transient digitizer, connected to the PMT via BNC cables at $50\,\Omega$ impedance, operates at a sampling rate of $20\,\mathrm{MHz}$ which equals a spatial resolution of $7.5\,\mathrm{m}$. For technical simplicity, the water vapor DIAL only operates with an analog channel as its focus is set on measurements inside the PBL. Only for the very weak signals of the temperature rotational Raman channel, an adjustable discriminator allows for simultaneous photon counting (FAST-MCS6).

  Regarding the retrieval of humidity profiles, much fundamental work has been published on the water vapor DIAL equation,

discussing major considerations that have to be respected (e.g. Schotland, 1974; Ansmann, 1985; Ansmann and Bosenberg, 1987; Bösenberg, 1998). Based on these detailed publications, and adapting the successful implementation as described in Vogelmann and Trickl (2008), the absolute water-vapor molecule concentration $N_{\mathrm{H_2O}}$ is calculated by using the well-known

DIAL equation as follows:

$$N_{\mathrm{H_2O}} = \frac{1}{\Delta\sigma_\uparrow(r) + \Delta\sigma_\downarrow(r)} \left[ -\frac{d}{dr} \ln \frac{S_{\mathrm{on}}(r)}{S_{\mathrm{off}}(r)} + G(r) \right]. \tag{1}$$

Here, $S_{\mathrm{on/off}}$ are the measured signals from the absorbed "online" wavelength $\lambda_{\mathrm{on}}$ and the unabsorbed "offline" wavelength $\lambda_{\mathrm{off}}$. $\Delta\sigma_\uparrow$ and $\Delta\sigma_\downarrow$ are the effective absorption cross sections of water vapor, respectively for the upward path of the narrow band laser light and the downward path of the spectrally Doppler-broadened, backscattered light from air molecules. $r$ is the vertical distance from the lidar and $G$ refers to a correction term which is needed in order to account for the spectral variation of the backscattered light which is molecularly Doppler-broadened (see Bösenberg, 1998). Therefore, this term is dependent on the water vapor distribution itself as well as the ratio between molecular and particle backscatter. As can be seen in Bösenberg (1998), this term can become relevant in two cases. The first case would be that the molecular backscatter coefficients $\beta_{\mathrm{M}}$ are much higher than the backscatter coefficients $\beta_{\mathrm{P}}$ from aerosols ($\beta_{\mathrm{M}}/\beta_{\mathrm{M+P}} \approx 1$) - which is usually not the case for observations within the PBL. The second case in which this term can become relevant is for atmospheric conditions in which there are strong gradients of $\beta_{\mathrm{M}}/\beta_{\mathrm{M+P}}$ - which is indeed the case within the PBL. This effect could principally be reinforced by the occurrence of fluorescing aerosol, caused by the emission of the 355 nm channel, which might act as a masked form of aerosol backscatter coefficients. In the current configuration of very little energies in the 355 nm channel ($\approx$ 1W), this effect is most probably negligible. Nevertheless, this effect has to be kept in mind in the case of substantially increasing pump lasers. However, after testing the effects of the correction term $G$, we found that for most heights of the PBL, the error is not dominant as it stays below values of $\approx$2%. At those altitudes, however, where there are strong gradients of $\beta_{\mathrm{M}}/\beta_{\mathrm{M+P}}$, the calculation of those is far from being trivial as the calculation of the gradient is very noisy if it is calculated from bin to bin. On the other hand, if the gradient is calculated over multiple bins, the locally very sharp aerosol gradients are artificially broadened and therefore inflict the corrections over a large range of altitudes. This bears the risk of applying a wrong correction at altitudes where $G$ shouldn't play a role. For those reasons, the herein presented humidity calculations omit the correction term $G$, following the reasoning that an overall small error is preferential over the introduction of artificial errors due to the implementation of a problematic correction term. Recent instrument developments show a method which could potentially circumvent this problem by adding a high-spectral-resolution lidar (HSRL) channel to DIAL systems, providing reliable information on molecular and particle backscatter properties without any dependence on a proper Klett inversion algorithm (Klett, 1985; Stillwell et al., 2020; Spuler et al., 2021; Hayman et al., 2024). Despite the fact that the ATMONSYS has been designed without an additional HSRL channel, following the thoughts of (Späth et al., 2020), effects of the Rayleigh-Doppler-broadened signal are minimized if the online frequency is chosen to be near the inflection point of the absorption line as done for this measurement set, taken on a humid summer day. In order to rule out that Rayleigh-Doppler broadening crosses the absorption line peak, we choose an absorption line under the premise of ensuring a suitable optical depth resulting in a related wavelength $\lambda_{\mathrm{on}}$ in a position further out than the actual inflection point of this absorption line. This, however, comes on the downside of not fully guaranteed minimization of a potential Rayleigh-Doppler error. The online frequency has been chosen to be $\lambda_{\mathrm{on,vac}} = 817.2460$ nm which is in the flank of an absorption line which centers at $\lambda_{0,\mathrm{vac}} = 817.2231$ nm (Ponsardin and Browell, 1997). The offline frequency has been set to $\lambda_{\mathrm{off,vac}} = 817.3526$ nm. Further details on the DIAL

specifications can also be read in Table 2. Beyond that, Vogelmann et al. (2022) provides a detailed description regarding resonator stabilization, seeding, and spectral characteristics of the laser beams. The calculation of the effective $\sigma_{\mathrm{on/off}}$ both in upward and downward direction is a crucial point within the DIAL equation. In order to account for the Lorentz-pressure-

broadening and the Rayleigh-Doppler-broadening, a convolution of those two effects has to be calculated which is described by a Voigt function. This function is dependent on both pressure and temperature over height. Therefore, in order to calculate precise absorption coefficients, additional information on the atmospheric state is needed. More details on both the calculation of the $\sigma$-profiles and the spectroscopic line characteristics can be found in Ponsardin and Browell (1997), Bösenberg (1998) or Vogelmann and Trickl (2008). For all data presented in this manuscript, the prevailing atmospheric pressure and temperature

conditions were taken from radiosonde ascents that are operationally performed by the DWD directly nearby (Chap. 3).

For the numerical inversion of the DIAL equation, Eq. 1 can be rearranged to

$$N_{\mathrm{H_2O}} = \frac{1}{\Delta\sigma_\uparrow(r) + \Delta\sigma_\downarrow(r)} [-\frac{1}{q(r)} \frac{d}{dr} q(r) + G(r)], \tag{2}$$

avoiding any asymmetric noise behavior caused by the logarithm and where $q(r) = \frac{S_{\mathrm{on}}(r)}{S_{\mathrm{off}}(r)}$. The numerical solution to the term $\frac{d}{dr} q(r)$ is implemented by calculating the slope of a least squares linear regression line. The regression line at one data point

is calculated by using symmetrically distributed neighboring data points. Following the explanations within Vogelmann and Trickl (2008), this finally leads to the numerical solution

$$N_{\mathrm{H_2O}}(r_{\mathrm{i}}) = \frac{1}{\Delta\sigma_\uparrow(r_{\mathrm{i}}) + \Delta\sigma_\downarrow(r_{\mathrm{i}})} [-\frac{1}{q(r_{\mathrm{i}})\delta_{\mathrm{i}}} \frac{\sum (j-i)q(r_{\mathrm{j}})}{\sum (j-i)^2} + G(r_{\mathrm{i}})], \tag{3}$$

where $i$ is the respective bin of interest, $k$ is the amount of neighboring points into one direction, $\sum = \sum\limits_{j=i-k}^{j=i+k}$, and $\delta_{\mathrm{i}} = r_{i+1} - r_{\mathrm{i}} = 7.5\,\mathrm{m}$. The total linear regression interval length is given aas $B = 2k + 1$.

The statistical uncertainty of lidar measurements is driven both by electronic noise and disturbing atmospheric noise. For DIAL systems, those statistical uncertainties are calculated by means of Gaussian error propagation which, in the end, leads to the uncertainty

$$\sigma_{N_{\mathrm{H_2O}}}(r_i) = \frac{1}{\Delta\sigma_\uparrow(r_i) + \Delta\sigma_\downarrow(r_i)} \cdot \frac{1}{q(r_i)} \cdot \sqrt{\frac{a(r_i)^2}{q(r_i)^2}\sigma_{\mathrm{q}}^2(r_i) + \sigma_{\mathrm{a}}^2(r_i)}, \quad \text{with} \quad a(r_i) = \frac{d}{dr} q(r_i). \tag{4}$$

Here, $\sigma_q(r_i) = \sqrt{\frac{1}{(2k+1-2)} \sum (a(r_i)r_j + b(r_i) - q(r_j))^2}$ and $\sigma_a(r_i) = \frac{\sigma_q(r_i)}{\delta_i \sqrt{\sum (j-i)^2}}$. $b_i = i\delta_i a(r_i) + \frac{1}{2k+1} \sum q(r_i)$ denotes the

axis intercept of the regression line mentioned above. As can be seen by the previous Eqs. 2/4, the total interval length $B$ influences both the spatial resolution of the measurement as well as the statistical uncertainty. In order to keep a reasonable signal-to-noise ratio (SNR) throughout the PBL, the DIAL retrieval decreases its spatial resolution towards higher altitudes with a step function $\propto r^2$ (Fig. 3). The effective vertical resolution of radar and lidar systems is defined by VDI (1999). According to Vogelmann and Trickl (2008), this equals to about a third of the interval width that is chosen for the derivation

of the logarithmic signal ratio in the DIAL equation. The parameters for the variable resolution are chosen with the intention

of keeping the standard deviation of the calculated water vapor concentrations pretty much constant over height and below 5 %. Due to its measurement principle, the DIAL relative uncertainty is scarcely dependent on the absolute values of specific humidity, further information on the statistical uncertainty of the ATMONSYS measurements will be shown in more detail in Sec. 4.2. During some measurement periods, the calculated humidity profiles show an odd artifact of too high concentrations at low levels. However, due to the DIAL principle and the instrumental setup, where both signals $\lambda_{on}$ and $\lambda_{off}$ take the same pathway which, as a consequence, should cancel out any identical overlap behavior within Eq. 1, this cannot be a classic overlap issue. Therefore, we assume that there has been an issue with a detector overload which leads to this artifact, despite the fact that the clip flag hasn't been set by the transient digitizer. However, as introduced at the beginning of this section, detector overflow could still have taken place without any clear notice. This assumption is supported by the fact that the issue with too high humidity values towards ground level is not identical at all times. As the reason for this behavior remains to be unclear, the presented data hasn't been modified by any correction function. Therefore, the absolute values below $\approx 0.5\,\text{km}$ above ground have to be interpreted with some caution. Their relative changes, however should not be affected by potential artifacts. The aerosol data shown in Sec. 4.2 results from the ATMONSYS elastic backscatter aerosol channel at $532\,\text{nm}$. This data has the same temporal resolution of $10\,\text{s}$ but a higher vertical resolution of $7.5\,\text{m}$. The aerosol backscatter coefficients are calculated with the Klett inversion algorithm (Klett, 1985; Speidel and Vogelmann, 2023).

**Table 2.** Water vapor DIAL parameter list.

| Parameter | Value |
| --- | --- |
| $\lambda_{on}$ | $817.2460\,\text{nm}$ |
| $\lambda_{off}$ | $817.3526\,\text{nm}$ |
| Spectral filter center wavelength | $817.2\,\text{nm}$ |
| Spectral filter bandwidth | $0.5\,\text{nm}$ |
| Online-offline switching rate | $100\,\text{Hz}$ |
| Sampling frequency | $10\,\text{s}$ |
| Bin width | $7.5\,\text{m}$ |
| Measurement range (Calculated for humid summer day conditions with realistic aerosol load) | $\approx 0.5$ - min. $3\,\text{km}$ |
| Effective vertical resolution (ground level - 3.5 km above ground) | $14$ - $214\,\text{m}$ |
| Statistical measurement uncertainty (Calculated for humid summer day conditions with realistic aerosol load) | $<5\,\%$ |
| Field of view | $2.5\,\text{mrad}$ |
| Full angle beam divergence | $<0.8\,\text{mrad}$ |

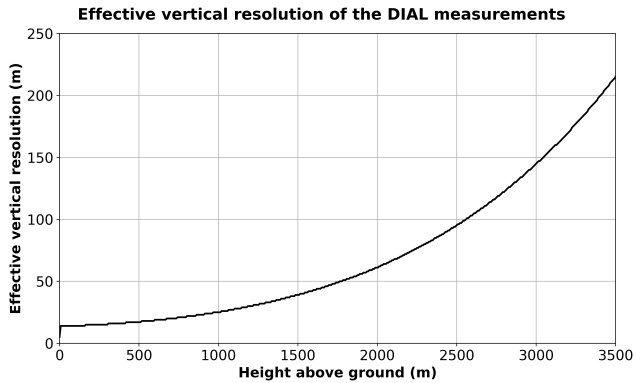

**Figure 3.** Effective vertical resolution for the absolute humidity calculated from the ATMONSYS DIAL.

## 3 The FESSTVaL campaign and complementary measurements

The Field Experiment on Submesoscale Spatio-Temporal Variability in Lindenberg (FESSTVaL) campaign has been carried out during summer 2021 in the north-eastern part of Germany (Hohenegger et al., 2023). The main objective of this campaign was to measure the submesoscale variability of the thermodynamic state of the PBL on a kilometer-scale. Therefore, a dense
network of in-situ and remote sensing detectors had been deployed within a $15\,\mathrm{km}$ radius around the Meteorological Observatory Lindenberg – Richard Aßmann Observatory (MOL-RAO) of DWD. This measuring effort was motivated by the need for enhanced data knowledge on such scales in order to derive and validate convection-resolving model parameterizations as already introduced in Sec. 1.

While many of the sensors were distributed across the observation area, a conglomeration of instruments was positioned di-
rectly at MOL-RAO which is situated on a small hill, overlooking the overall flat surrounding terrain. For the ATMONSYS DIAL, this gave the rare opportunity of both comparing different humidity lidars with each other and with the operational radiosondes at MOL-RAO. In addition, a co-located Doppler wind lidar allowed for combined measurements in order to calculate vertical fluxes of latent heat. The vertical staring Doppler wind lidar was installed directly next to the ATMONSYS container. The local distribution of all instruments used within this manuscript can be seen in Fig. 4. In the following, every instrument
that has been used for comparison or combined calculations is briefly described:

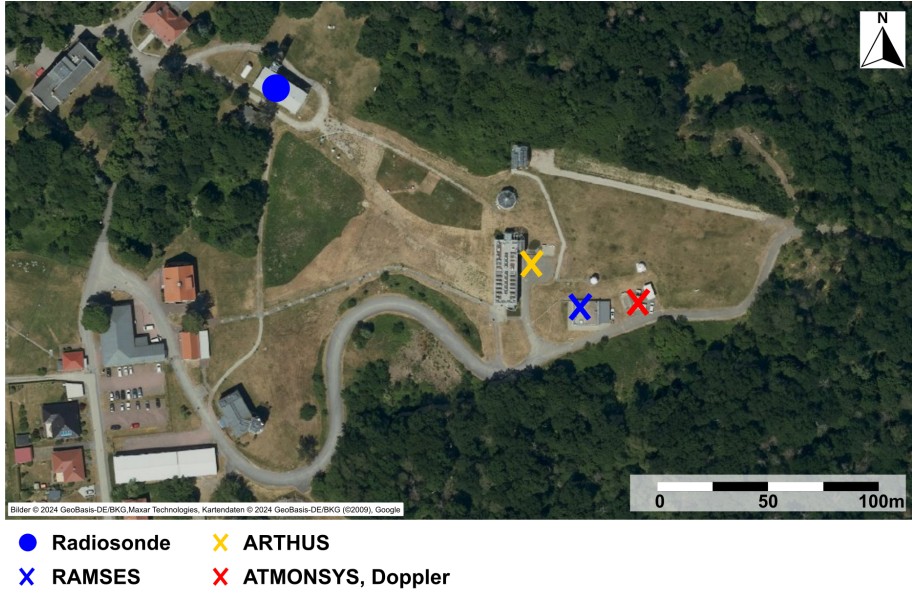

**Figure 4.** Overview of the measurement site and the spatial distribution of instruments during FESSTVaL at MOL-RAO in Lindenberg.

**RAMSES lidar**

At Lindenberg, DWD operates the autonomous Raman lidar for atmospheric moisture sensing (RAMSES) since 2005. After several extensions, RAMSES has evolved into a spectrometric fluorescence and Raman lidar with several receiver units, including three spectrometers. Unfortunately, due to maintenance on its air conditioning system, RAMSES did not operate
continuously during the ATMONSYS deployment in Lindenberg. However, it was possible to have a small temporal overlap in which both systems operated, which is sufficient to allow for a comparison of the measured data. For the case study presented here, data from the discrete detection channels of the near- and far-range receivers (Reichardt et al., 2012), and of the UVA spectrometer (Reichardt et al., 2023), which is a subsystem of the near-range receiver, are analyzed. Because of its complex receiving system, RAMSES possesses duplicate measurement capability for most measured quantities. For instance,
to obtain optimum measurements of water vapor mixing ratio, data from the near-range receiver (discrete detection channels before and UVA spectrometer after sunset, respectively) below 2 km and the far-range receiver (discrete detection channels) above are merged. In contrast, particle backscatter coefficient and depolarization ratio are calculated using only data of the discrete detection channels in the far-range receiver. The fluorescence backscatter coefficient in the cyan wavelength range is determined by integrating the fluorescence spectrum measured with the UVA spectrometer (nighttime operation only) between
455 to 535 nm. In this study, 240 s of lidar data is integrated for each profile. The vertical resolution of the raw data is 60 m, and signal profiles are smoothed with a sliding-average length of 3, and 5 height bins between 1 and 2.5 km, and above 2.5 km, respectively. With these settings, statistical measurement errors of the water vapor mixing ratio are typically between 5 % and 15 %.

**ARTHUS lidar**

Towards the end of FESSTVaL, an additional water vapor Raman lidar was set up at Lindenberg. The Atmospheric Raman Temperature and Humidity Sounder (ARTHUS) lidar system (Lange et al., 2019) is the non-commercial precursor of the Raman lidars from *Purple Pulse Lidar Systems S.L.*. It measures water vapor throughout the PBL during nighttime and daytime conditions at a sampling rate of at least 10 s. Although it has not been directly involved in the FESSTVaL campaign, we could benefit from the synchronous measurement. This gives us the advantage of an intercomparison between all three water vapor

lidar systems. The ARTHUS data has been obtained with a vertical sampling of 3.75 m at a temporal resolution of 10 s. The signal has been smoothed by a vertical sliding average of 26 bins, resulting in an effective vertical resolution of ≈100 m. The statistical uncertainty of the ARTHUS measurements is determined by applying error propagation to the measured photon counts (or virtual photon counts for analog signals) (Wulfmeyer et al., 2016). This method accounts for shot noise, the primary source of error in Raman lidar signals. To obtain the total statistical uncertainty, an autocorrelation analysis is performed on

a time series of the measured parameter fluctuations (Behrendt et al., 2015; Lenschow et al., 2000). This approach distinguishes between uncorrelated noise and correlated atmospheric fluctuations, enabling the simultaneous retrieval of total noise uncertainty profiles and higher-order moment profiles of atmospheric fluctuations, along with their associated uncertainties.

**Doppler wind lidar**

Measurements of vertical fluxes with lidar systems depend on the combination of vertical thermodynamic profiles with vertical

wind information. Therefore, vertical staring Doppler wind lidars with a very high temporal resolution are needed. At MOL-RAO, we operated a Streamline XR Doppler lidar (Halo Photonics). The instrument has a range gate length, and therefore vertical resolution, of 48 m while using a maximum of 125 range gates which, under good meteorological conditions, leads to a maximum vertical range of 6000 m. The Doppler lidar measured with 20,000 pulses per ray and a sampling rate of ≈3 s. The system has a pulse width of 330 ns, a pulse repetition frequency of 10 kHz, and a wind velocity statistical measurement

uncertainty of 0.1 m/s[1]. We removed the data with a high noise level by filtering with a relatively low Signal-to-Noise Ratio (SNR) + 1 threshold of 1.000 to keep the data availability high (as in Wolz et al., 2024). The SNR values are used as a quality indicator of the radial velocity measurements and are generally output by the Doppler lidar. The system calculates the SNR values by comparing the sensor's internal noise level with the intensity of the backscattered light from the system's emitted pulsed laser beams. As the spatial variability inside the PBL is known to be quite high, we situated the Doppler lidar as close

as possible to the ATMONSYS DIAL at a horizontal distance of < 5 m. By this, both lidars should measure mostly the same air volume, with small discrepancies due to differing beam divergence. This setup has been realized only for a selected time period as the Doppler wind lidar was operated at different sites and in different scan configurations during the campaign.

**Radiosonde**

At MOL-RAO, DWD conducts four daily routine ascents of Vaisala RS41-SGP radiosondes per day. During the time of

FESSTVaL, additional ascents have been conducted in cases of promising atmospheric conditions regarding the campaign

goals of FESSTVaL. The release site of the radiosondes was in close proximity to the ATMONSYS container within a distance of $\approx 200\,\mathrm{m}$. Detailed information on the technical specifications of the radiosondes is given directly by the manufacturer (https://docs.vaisala.com/v/u/B211444EN-J/en-US). For measurements of relative humidity, a sounding uncertainty of $4\,\%$, and temperature-dependent response times of $<0.3\,\mathrm{s}$ ($20°\,\mathrm{C}$) and $<10\,\mathrm{s}$ ($-40°\,\mathrm{C}$) are claimed. This data is available for the entire time period in which ATMONSYS was operating (Hohenegger et al., 2023).

**Measurement day: 18 July 2021**

All data presented in this manuscript have been collected during 18 July 2021. This specific day has been chosen for two reasons. First, it has the advantage that all of the above introduced instruments were operating simultaneously and at the very same location. Second, the prevailing weather on this day led to representative conditions of a typical convective summer day. Fig. 5 gives a broad overview of the atmospheric conditions and development during the second half of this specific day. This overview graphic has been generated with data from the continuously running RAMSES lidar at a temporal resolution of $4\,\mathrm{min}$. Already at this temporal resolution, frequent changes between high and low humidity concentrations within the lowest $2\,\mathrm{km}$ reveal convective behavior (Fig. 5a). Also, a typical moistening of the PBL during day can be observed with a simultaneous increase of the PBL height - visually defined by strong vertical humidity gradients, vertical gradients in the particle backscatter coefficients (Fig. 5b), and partly by the particle depolarization ratio (Fig. 5c). However, it has to be stressed that the determination of the PBL height is somehow tricky and strongly dependent on the method being used (e.g. Foskinis et al., 2024; Kotthaus et al., 2023, and references therein). Definitions of aerosol and humidity gradient based PBL heights do not necessarily coincide with thermodynamic or kinetic energy definitions of the PBL top. Therefore, as additional information, data from the available radiosonde ascents have been used to calculate the PBL top (Tab. 3) defined by the bulk Richardson number as described in Seibert et al. (2000) and Zhang et al. (2022). At least for high noon, those values show to be in good agreement with the visual perception of humidity and aerosol backscatter gradients. A Doppler wind lidar at a nearby site was measuring horizontal wind speeds by doing velocity-azimuth display (VAD) scans. Based on the measured values for the horizontal wind speed maximum, showing the nose of the low level jet (and therefore the PBL height) the PBL top can be validated to be at altitudes of $1500\,\mathrm{m}$ AGL during high noon.

**Table 3.** PBL height values for 18 July 2021, defined by the lowest altitude where bulk Richardson numbers $Ri_\mathrm{B} > 0.25$. The calculations base on data from radiosonde ascents at MOL-RAO Lindenberg.

| Time of the radiosonde ascent [UTC] | Boundary layer top [m AGL] |
| --- | --- |
| 10.75 | 1530 |
| 16.75 | 1200 |
| 22.79 | 300 |

Besides a general idea of the PBL development, the particle backscatter coefficient shows the development of convective clouds passing over the lidar from 12 UTC to 16 UTC. Starting around 20 UTC, dense clouds develop underneath an elevated aerosol layer which is located at $\approx 2.5$ km. Furthermore, the combination of Figs. 5b-d reveals valuable information about the dynamical situation of aerosol concentrations during the second half of this day. From 12 UTC until 20 UTC, the particle depolarization ratio shows increased values between $\approx 2$ -4 km, hinting towards elevated concentrations of dust particles. Be-
tween 20-24 UTC, the already mentioned enhanced particle backscatter coefficients at $\approx 2.5$ - 3.5 km show almost no particle depolarization ratio. However, the fluorescence backscatter coefficient of this layer is significant with very high vertical gradients. Based on those features, the origin of this aerosol layer can be connected with wildfires. Section 2.3 already discussed the potential issue with fluorescence backscatter in the 817 nm DIAL channel which could be caused by strong emission power at 355 nm. Although the ATMONSYS system does not operate with such high powers at this wavelength, Fig. 5d impressively
shows very steep fluorescent backscatter coefficient gradients.

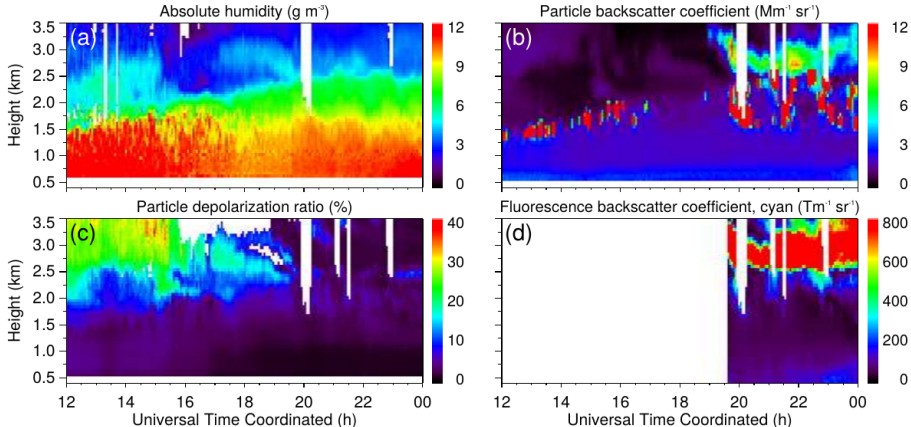

**Figure 5.** Overview panel of the atmospheric conditions on the second half of 18 July 2021 based on RAMSES data showing (a) absolute humidity, (b) particle backscatter coefficient, (c) particle depolarization ratio, and (d) fluorescence backscatter coefficient in the cyan wavelength range. Absolute humidity and fluorescence backscatter coefficient are calculated from photon-counting signals, particle backscatter coefficient and depolarization ratio from analog signals. For each profile, 240 s of lidar data is integrated, and the calculation step width is 120 s. The vertical resolution of the raw data is 60 m, and signal profiles are smoothed with a sliding-average length of 3, and 5 height bins between 1 and 2.5 km, and above 2.5 km, respectively. White areas indicate where data were missing or rejected by the automated quality control process. The local apparent solar noon on that day is shortly after 11 UTC.

## 4    Data evaluation of the ATMONSYS DIAL

### 4.1    General instrument performance

All ATMONSYS DIAL data presented within this publication has been recorded at a sampling rate of 10 s. This is the highest temporal resolution in which the system has been operated so far. Though somehow arbitrary, this value is a trade-off decision

in order to allow for observations of turbulent transport processes while, at the same time, keeping a good SNR and, therefore, reliable measurements over a high range of altitude. With the very same intention towards signal quality and vertical range, the DIAL retrieval decreases its spatial resolution as already shown in Sec. 2.3.

On 18 July 2021, this resulted in a maximum vertical range for the water vapor DIAL of $\approx 3.2\,\mathrm{km}$ under clear sky conditions even around the time of the daily apex of the sun's motion (Fig. 6a). This maximum range value naturally changes mostly with the prevailing atmospheric conditions. In this particular case it can be seen that the $3.2\,\mathrm{km}$ correspond to an altitude with a thin layer of nearly zero aerosol concentration. The above lying aerosol layer with slightly higher particle backscatter coefficients, at least in the presented case, does not lead to a reasonable DIAL signal anymore (Fig.6b). Furthermore, at an altitude between $\approx 2\text{-}2.5\,\mathrm{km}$, Fig. 6b shows a very significant wave-pattern which is not that apparent within Fig. 6a. The reasons for this "wave-like" structure are not clearly attributable. The presented time period has been chosen as it is (mostly) free of small convective clouds and constantly allows for smooth values over the full height. Regardless of the fact that there is enhanced aerosol up to an height of at least $3.2\,\mathrm{km}$, it has to be stressed that the PBL top is situated much lower in this specific case, as already indicated by Tab. 3. The application of an automated boundary layer height detection algorithm to the aerosol data similar to Baars et al. (2008) determines the PBL top to a height of $\approx 1.2\,\mathrm{km}$. This altitude of steepest gradients can easily be confirmed by eye for both aerosol and humidity distribution. However, as discussed earlier, a humidity/aerosol gradient-based estimation of the PBL top does not necessarily coincide with its thermodynamic definition leading to differing numbers in comparison to Tab. 3.

Over the presented time period of $1\,\mathrm{h}$, structures of alternating moisture concentrations can clearly be observed throughout the entire PBL in the form of darker plumes (e.g. at $\approx 11.8\,\mathrm{UTC}$) which are surrounded by drier time periods with lighter color shading. Those changes can also be studied more detailed within the afterwards described Fig. 7a. As the structure of those humidity plumes is non-periodic, we can address this behavior to convection. However, the direct linkage between convective updrafts and changes in humidity is not always straightforward as can be seen in the following. For an exemplary purpose, in order to visualize correlations in changes of humidity and aerosol concentration with convective phases, their changing structures can be highlighted by calculating absolute and relative changes to the median in each height level:

$$\Delta x_{rel} = \left(\frac{x}{\tilde{x}}\right) - 1,$$
$$\Delta x_{abs} = x - \tilde{x}. \tag{5}$$

Here, $x$ can be either absolute humidity or aerosol backscatter coefficients. $\tilde{x}$ is the median of the respective measure at each single height level over the entire time period that is plotted in Fig. 6. In contrast, to directly show the actual vertical movement of the air parcels, Fig. 6c visualizes the measured absolute vertical wind velocity. By this formula, positive values show an overall increase of concentrations compared to the prevailing median and vice versa for negative values. These changes (Fig. 7a/b) can be compared with the prevailing vertical winds measured by the Doppler wind lidar positioned directly next to the ATMONSYS system (Fig. 7c). For the changes in humidity, one has to keep in mind that the absolute values of humidity as observed in Fig. 6a drop above the PBL. This allows for several insights. First, the changes of absolute humidity (Fig. 7a)

are slightly larger above the PBL than within the PBL, whereas the relative changes are substantially larger above the PBL. To our understanding, this can be attributed to turbulent mixing inside the PBL which becomes apparent in constant vertical humidity concentrations inside the PBL (e.g. Stull, 1988; Couvreux et al., 2005; Muppa et al., 2016). Above the PBL, on the other hand, lacking turbulence and entrainment processes lead to much more horizontal heterogeneity which is advected by free tropospheric winds and therefore leading to larger relative changes of moisture concentrations over time. At the same time, a statistical SNR feature cannot be ruled out as the absolute humidity concentrations above the PBL become lower which can increase the relative deviation as well. Nevertheless, the behavior of increasing relative moisture variability over height is apparent in other observations as well (e.g. Van Baelen and Penide, 2009; Vogelmann et al., 2015; Hicks-Jalali et al., 2020), however at much lower temporal resolution. This can at least partially be attributed to the fact that upper edge of the PBL is a mixing layer, where turbulent mixing of humid air from the PBL and dry air from the free troposphere takes place. This potentially results in a very heterogeneous water vapor distribution in this altitude region, which streams through the probed volume with the synoptic wind. The attribution of vertical variability will be discussed in more detail in Sec. 4.2. Second, we can ascertain that the correlation between moisture and aerosol concentration on the sub-minute timescale (Figs. 7a and 7b) is quite complex and not at all straightforward. As can be seen e.g. around 12 UTC and 12.3 UTC and especially around the PBL top, there are times in which high moisture values coincide with high aerosol concentrations and vice versa for low values. This could be explained by a common source of aerosol particles and moisture at earth's surface. Also, although there is a lot of noise in the moisture changes above the PBL, one can interpret quite similar sinusoidal patterns for both moisture and aerosol concentration. A simple temporal moving average for the humidity data, which is not included here, does denoise the patterns. Nevertheless, this doesn't change the general picture of some identical patterns which are, however, not always temporally and spatially consistent with the aerosol pattern. Therefore, a general and straightforward correlation between aerosol and humidity changes is not apparent. The same observation can be stated if one includes the vertical winds (Fig. 7c). Again, there are phases in which e.g. the sinusoidal pattern of the aerosol changes at $\approx$ 1.8-2.3 km above ground level (AGL) coincide with respective up-winds and down-winds. However, the connections are not always obvious. Finally, the very same observation is also true if one compares the changes of humidity with vertical wind speed. Again, it can be seen by eye, that enhanced moisture concentrations are not always coinciding with a certain direction of vertical motion. This shows that the horizontal advection by large eddies and mean horizontal wind plays a major role and has to be considered. Especially above the PBL, we consider horizontal advection as the most dominant source of inhomogeneity which should lead to a certain decoupling of moisture and aerosol changes to the vertical wind velocity. As can be seen, the combination of data in Fig. 7 contains a lot of potential. We expect that an in-depth analysis of correlations and causalities between humidity, aerosol concentration and vertical winds over multiple days is of high interest in terms of convection initiation and cloud formation. However, the much more detailed analysis of potential coherence between those measures is beyond the focus of this manuscript.

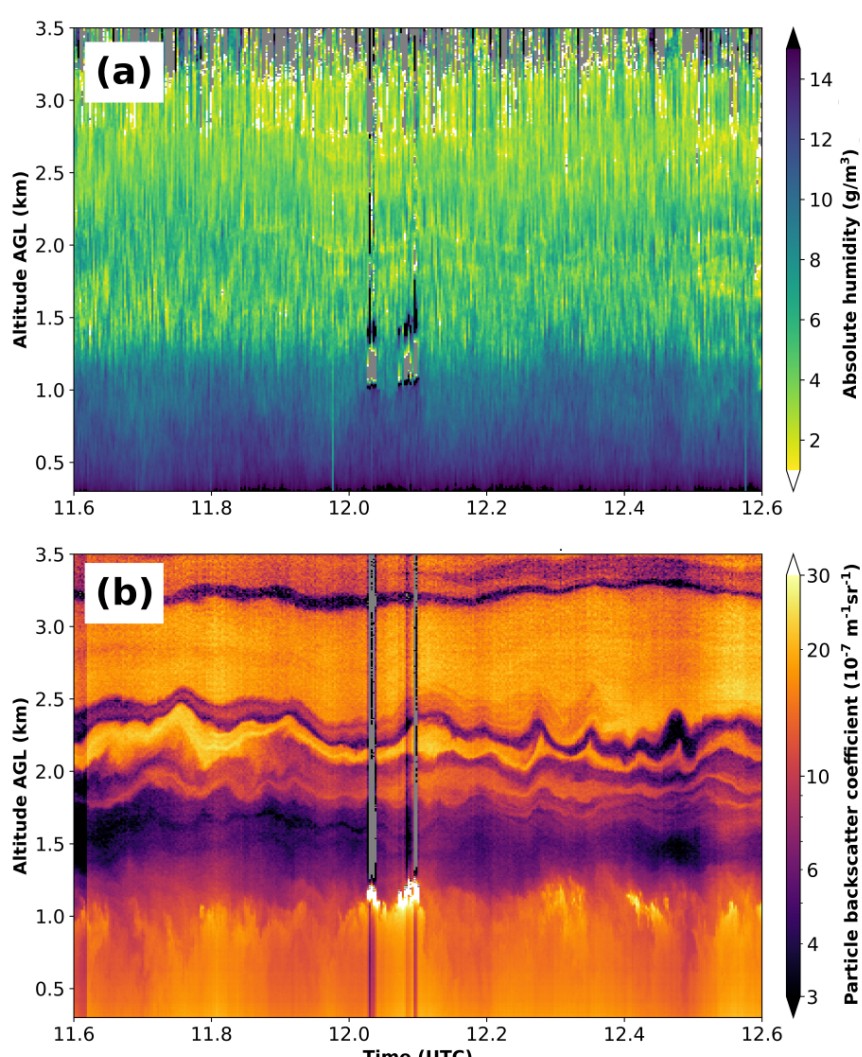

**Figure 6.** ATMONSYS lidar data showcase from 18 July 2021 for vertical and temporal highly resolved time series of (a) absolute humidity and (b) aerosol backscatter coefficients during high noon under clear-sky conditions and a temporal resolution of $\Delta t = 10\,\text{s}$. No temporal smoothing has been applied.

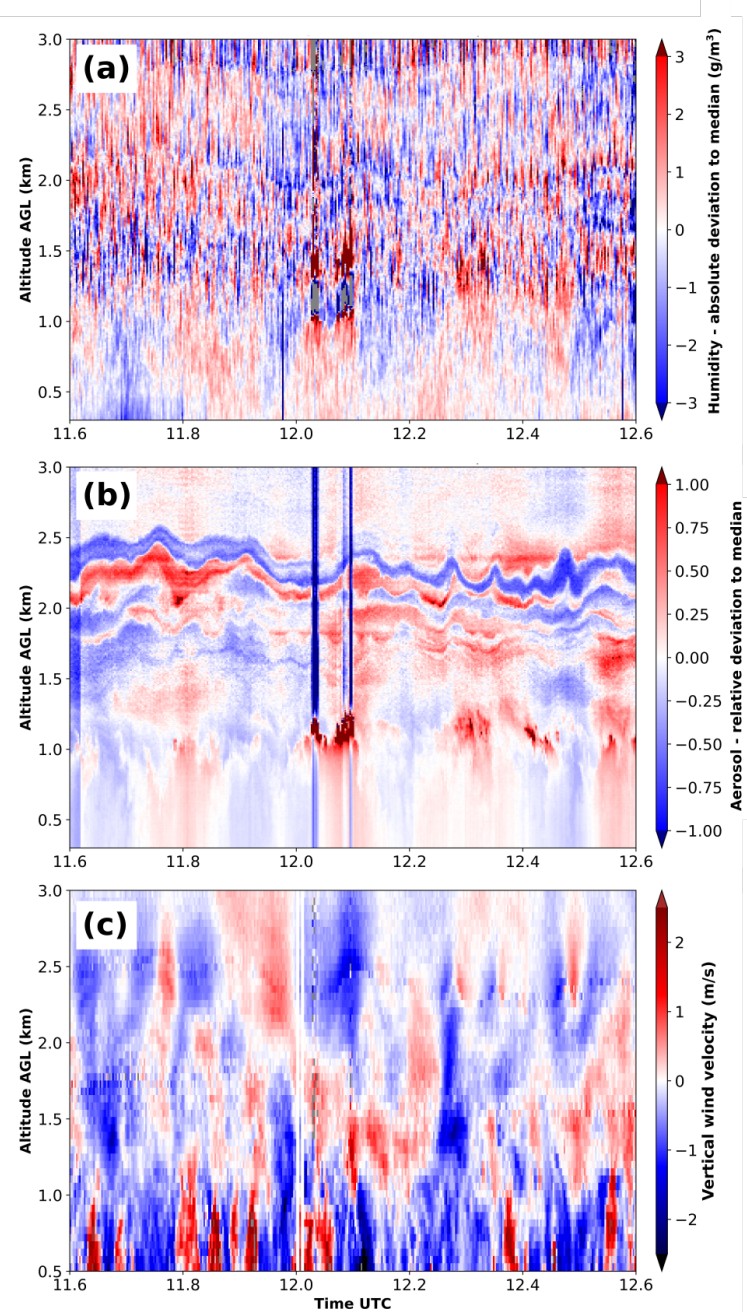

**Figure 7.** Showcase from 18 July 2021 for vertical time series of deviations from the median at each height level (Eq. 5): (a) absolute deviations of the absolute humidity and (b) relative deviations of aerosol backscatter coefficients. Vertical disruptions in aerosol backscatter coefficients are caused by missing reference values in the far end (Klett inversion) during cloudy conditions. The absolute vertical wind velocity, as measured by the Doppler wind lidar is shown in (c). All measurements show a time period during high noon under clear-sky conditions with a temporal resolution of $\Delta t = 10\,\mathrm{s}$. No temporal smoothing has been applied.

## 4.2 Uncertainty and variability

In the previous section, Fig. 7(a) already visualized the variability of calculated absolute humidity values over height and time.
The amplitude of relative deviations to the median value on each respective altitude showed to be considerably lower inside the PBL and much higher in the lower free troposphere. However, purely based on the previously discussed graphics (Figs. 6,7), it is unclear to address the reasons for this pattern as several causalities are plausible. On the one hand, decreasing signal strengths from higher altitudes proportional to $1/r^2$ and overall lower humidity concentrations are leading to much lower SNR values and therefore higher statistical noise. This physical restriction is reflected in an increased profile-to-profile variance which can be misinterpreted as atmospheric variability. This issue, however, can be antagonized by increasing the spatial interval length for the linear regression inside the DIAL algorithm (Eq. 1), leading to lower spatial resolution but also lower statistical noise. On the other hand, changes in concentrations of water vapor in the lower free troposphere are much more related to horizontal advection by substantially stronger winds as they are inside the PBL. Different sources of origin and less turbulent mixing can therefore lead to substantial differences of atmospheric humidity concentrations rolling by the lidar's vertical measuring column. Therefore, it is not possible to directly attribute the observed deviation fluctuations to issues with signal strength or actual atmospheric conditions. In consequence, a quantification of uncertainties based on all single 10 s profiles from the very same time period as before (11.6 - 12.6 UTC, $\approx$ 360 profiles) is presented in the following (Fig. 8). For this quantification of uncertainties, dense cloud-inflicted profiles that show unrealistic, strongly oscillating values are ignored above the clouds and therefore not included in the statistics. They are detected based on their simple feature of reaching negative values at an altitude in which humidity is far away from zero. The negative values occur due to the fact that, in the event of clouds in the laser pathway, the stronger signal for $\lambda_{off}$ reaches a saturated non-linear response first near the cloud base. As a consequence, afterwards, $\frac{d}{dr} \ln \frac{S_{on}(r)}{S_{off}(r)}$ is changing its sign above the cloud base due to a further increase of $\lambda_{on}$, leading to negative values of the retrieved water vapor concentration. Accordingly, the filtering is done in the simple way that we look for the respective, cloud-influenced profiles that reach values less than zero within the lowest 3 km. This proved to be a straightforward and at the same time reliable method, at least under the weather conditions at the instrument site for the presented data. The spread of values for all single profiles within the respective time period is presented by the dashed gray lines in Fig. 8(a). At first glance, it is obvious that the spread of values above $\approx 3.2$ km is too high for reliable humidity values. This has already been seen directly in the time series before (Fig. 6). Below that altitude, the spread is reduced and reveals several characteristics. For a better visualization of the majority of measurements, the gray shaded area shows all values between the p10 and p90 percentiles. By this, single extreme outliers which can be caused by atmospheric disturbances (e.g. insects, condensation) or lowered laser performance (e.g. mode hop / reset of the seeding device) do not influence the overall distribution. The absolute width between p90 and p10 is visualized on the right side of the same graphic (Fig. 8b, gray line). According to our expectations, the spread of values is the lowest for the lowermost altitudes which is at least partially fostered by a high SNR. Above the PBL, especially at $\approx 1.3 - 2$ km, the calculated values show a larger spread than below. Interestingly, the p90-p10 width reduces again at around $\approx 2.5$ km, almost reaching the values from the lowermost measurement altitude. This can be understood as a direct proof that at least a significant portion of the wider spread of humidity concentrations at altitude levels of $\approx 1.3 - 2$ km is caused by actual

atmospheric variability and not by a decreasing SNR. This argumentation is reinforced by the relative standard deviation, which is directly calculated out of the linear regression during the DIAL retrieval (Vogelmann et al., 2015). The respective blue line in Fig. 8(b) shows to be relatively constant over height, always staying below values of 5% relative standard deviation. As already

described, this behavior is due to the dynamic spatial resolution over height (Sec. 2.3). It shows that, with this configuration, the ATMONSYS DIAL stays within the before mentioned criteria for observation uncertainties $< 5\,\%$. In addition to the standard deviation that can be determined from the linear regression inside the retrieval algorithm, Fig. 8(b) also shows the profile-to-profile relative standard deviation (red line). This means that the standard deviation is calculated with respect to the mean of all values inside the selected time period.

What can be seen directly from both the red and the blue line is, that there is a maximum of values at $\approx 1.5$ - $1.7\,\mathrm{km}$. This is again an indication for the position of the PBL top. As dry air parcels from the free troposphere are entrained into the moist PBL and vice versa, the variance is the highest in the region of the PBL top, although the relative standard deviations lead to maxima at higher altitudes compared with the strong gradient of humidity. From the comparison of the profile-to-profile standard deviation (Fig. 8(b), red line) with the standard deviation of the DIAL algorithm (Fig. 8(b), blue line) we can deduce

information on the spatial scale of the humidity variations. This is enabled by the fact that the prevailing moisture concentration within a chosen interval length $\mathrm{d}r$ defines the slope of the term $\frac{d}{dr} \ln \frac{S_{\mathrm{on}}(r)}{S_{\mathrm{off}}(r)}$ (Eq.1). Therefore, under the assumption of constant humidity throughout this interval length, the slope of this term would be constant and, hence, the standard deviation of the linear regression would be low. As a consequence, if the interval length is large enough to include different humidity regimes, the standard deviation of the DIAL algorithm increases and, thus, can be misinterpreted as statistical uncertainty. As the integration

length of the DIAL retrieval algorithm in this region is about $150\,\mathrm{m}$ we conclude on humidity variations on a smaller scale. A higher level of standard deviation in the DIAL algorithm might also be induced due to less humidity and therefore lower values of $\frac{d}{dr} \ln \frac{S_{\mathrm{on}}(r)}{S_{\mathrm{off}}(r)}$ or low SNR values. Nevertheless, this influence is not dominant as the standard deviation decreases towards higher altitudes albeit a further decrease in humidity. Above this layer, the values drop for both uncertainty estimators, similar to the before described p90-p10 width (grey line), but stay at higher levels than inside the PBL. From the difference between the

blue and the red line (Fig. 8(b)), we can also give an estimation of the atmospheric variability, which is related to the residual of the measured variations after subtraction of the instrumental uncertainties. Within the PBL, it covers values between 5% at the lowermost altitudes that grow with height up to $\approx 30\%$ in the area of the PBL top. This increase in variability with height shows that both horizontal advection by eddies and vertical entrainment into the PBL leads to much higher humidity variability than is generated by the ground-dependent heterogeneity of evaporation at the surface level. Both the magnitude of

variance values as well as the described behavior of increasing values over height can be confirmed by humidity measurements on several levels on very high towers (e.g. the Park Falls tower - data available at https://flux.aos.wisc.edu/fluxdata). Above the boundary layer top, the atmospheric relative variability shows to be at values around $\approx 20\%$ and more. Again, above $3\,\mathrm{km}$ the backscatter from aerosols diminishes which leads to very high uncertainties that, in contrast to lower altitudes, have to be addressed to instrumental noise caused by low SNR values.

A first impression of how the absolute values and their median from that time period compare to the in-situ RS41 radiosonde measurements is given by the red line in Fig. 8(a). This is, however, only a coarse intercomparison as the starting time of the

radiosonde has been at 10.75 UTC and only needs about 10 min to reach a height level of 3 km above ground. Additionally, the radiosonde is drifting with horizontal winds and is not probing the same volume. Nevertheless, the general vertical humidity distribution is still well represented by the sonde as there hasn't been any major meteorological changes within the time

difference of less than an hour. As already mentioned in the previous section, the time period around the ascent of the radiosonde is afflicted by sporadic occurrences of shreds of clouds. Nevertheless, Sec. 5.1 will show an intercomparison between cloud-free radiosonde profiles with the lidar measurements at the best possible temporal overlaps. The absolute deviations between the humidity, measured by the radiosonde and DIAL, especially within the PBL will be discussed there in more detail.

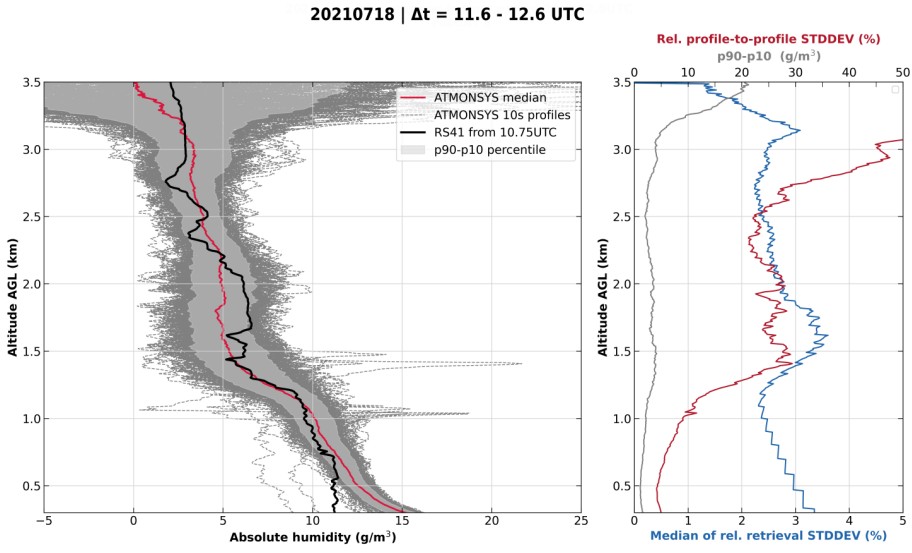

**Figure 8.** Analysis of absolute humidity profiles for the time period 11.6-12.6 UTC on 18 July 2021. (a): Single profiles (dashed lines) and their median value (red line) are compared with the RS41 radiosonde measurement ($t_0 = 10.75$ UTC, black line). The shaded area shows the p90-p10 percentile range. (b): The uncertainties are quantified by the relative standard deviation (STDDEV) - directly calculated out of the linear regression in the DIAL algorithm (blue line), the absolute width of the p90-p10 range (gray line), and the relative profile-to-profile STDDEV of all profiles in the time period (red line). Erroneous profiles above dense clouds are filtered and not considered in the shown profiles and statistics.

### 4.3    Turbulence spectra

The PBL is the atmospheric layer that is directly influenced by the earth's surface. Thermal convection as well as surface-induced friction causes both vertical and horizontal wind shear which in consequence leads to the formation of eddies and, therefore, turbulence. From a starting point, these eddies are driven by stronger winds in the free troposphere which, as a consequence of the friction, lead to eddies with big diameters that can even equal the PBL height. Depending on the apparent season, local weather conditions, daytime, as well as surface properties, the PBL height can vary from $< 100$ m up to $\approx 3$ km.

For the specific time period of investigation, judging from strong humidity and aerosol gradients as well as wind and bulk Richardson number (Sec. 3), the PBL height is $> \approx 1.2$ -$1.5$ km. Over time, these big eddies decay into multiple smaller ones

with diameters less than $1\,\text{m}$, leading to eddies of multiple diameters (e.g. Foken, 2017). The decay from larger to smaller eddies means a simultaneous energy decline. Inside the PBL and within isotropic turbulence conditions ($\approx 0.01 - 5\,\text{Hz}$), this decline of energy follows the so-called "Kolmogorov -5/3 law" (Kolmogorov, 1991). This law defines that a decline of energy density by the factor 5 is directly intertwined with an increase in frequency by the factor 3. The applicability of this law to real-world measurements has been demonstrated for a long time by multiple studies (e.g. Stull, 1988), however only rarely for humidity lidar measurements. Throughout the last three decades, a small collection of such turbulence investigations developed with DIAL measurements of, at that time, lower spatiotemporal resolution (Senff et al., 1994), methodological focus on the treatment of statistical high order moments for high-resolution DIAL measurements (Lenschow et al., 2000), and high-resolution Raman humidity lidars (Wulfmeyer et al., 2024).

Following this theorem, instruments can be tested on their capability of resolving turbulence which is frequently done (e.g. Moncrieff et al., 1997; Fratini et al., 2012; Brugger et al., 2016; Mauder et al., 2020a). In the logic of this theory, only within those spectral ranges where the turbulence spectra of the instrument show the -5/3 correlation, the measuring instrument is capable of resolving turbulence. The application of this theory is bound to the assumption of isotropic turbulence and only refers to the inertial sub-range. The methodology of this procedure is described e.g. by (Stull, 1988) and is based on the Fourier-transformed turbulence spectra of the measurements. For turbulence spectra, the Fourier-transformed always refers to the fluctuation of the mean value according to the Reynolds decomposition. This means that every measured value, here in the case of absolute humidity $q$, can be decomposed into the mean value $\overline{q}$ and its fluctuation $q'$ in the following manner: $q = \overline{q} + q'$. In the following, we present turbulence spectra over a time interval of $2\,\text{h}$ for the ATMONSYS DIAL humidity measurements $q'$ and for the Doppler wind lidar standing next to it, delivering vertical wind values $w'$ ($w = \overline{w} + w'$). In addition, the co-spectra from vertical humidity and wind measurements are shown as well. For the co-spectra, one has to keep in mind that the Kolmogorov-dependency for the energy decay in the inertial sub-range changes from -5/3 to -7/3 (e.g. Kaimal and Finnigan, 1994). In order to get reliable results from those co-spectra, the time series of measurements have to be carefully prepared so that they match with each other in the best possible way. As already described, the DIAL measurements have a temporal resolution of $10\,\text{s}$. The Doppler wind lidar was running with a higher temporal resolution of $\Delta t = 3\,\text{s}$. In order to have similar resolutions, the wind data has been block averaged to the identical resolution of $\Delta t = 10\,\text{s}$. Outliers from both time series are removed in an iterative way in which the standard deviation $\sigma$ is calculated individually for each height level of the time series. In a next step, all median-subtracted values that exceed a range of $5\sigma$ are removed. Then, the next iteration starts with a newly calculated median and $\sigma$. This is done until no data points are removed anymore. Removed values are replaced by interpolated values. If the amount of outliers exceeds a threshold of $10\%$, the entire time series is neglected. The time series is detrended by the application of a linear regression. Finally, the data is high-pass filtered with a boundary frequency of $30\,\text{min}$.

The comparison of the turbulence spectra for both humidity $q$ and vertical wind $w$ (Fig. 9) reveals several features. First, the wind spectra for $w'$ visually show to be in good agreement with Kolmogorov's law from $\approx 1.5 - 5 \cdot 10^{-2}\,\text{Hz}$ at all presented heights. At the frequency of $\approx 1.5 \cdot 10^{-2}\,\text{Hz}$, a drop towards lower energies is visible at all heights - although the spectra still show parallel behavior to the dashed Kolmogorov line down to $\approx 6 \cdot 10^{-3}\,\text{Hz}$. For all three altitudes, but most dominantly for the spectrum line at $500\,\text{m}$, a second drop of energies towards lower energy can be observed at $\approx 9 \cdot 10^{-3}\,\text{Hz}$. Similar,

but not identical structures can be observed for the humidity spectra. Here, agreement to the Kolmogorov spectrum is obvious in the frequency range of $\approx 2 - 3 \cdot 10^{-2}$ Hz across all altitudes. However, the spectrum of $q'$ does not always follow the dashed Kolmogorov line. Horizontal "energy plateaus" can be seen almost simultaneously at all three altitudes as e.g. from

$\approx 1.5 - 2 \cdot 10^{-2}$ Hz and $\approx 3 - 4 \cdot 10^{-2}$ Hz. Nevertheless, both at lower and higher frequencies, the spectra change again from plateaus to a decline of energy, parallel to the dashed line of the Kolmogorov spectrum. Even at the highest frequencies of the DIAL ($\approx 4 - 5 \cdot 10^{-2}$ Hz), where white noise would be expected the most, the spectra at all three altitudes show to be parallel to the Kolmogorov line. Interestingly, at least the $q$-spectra for $358$ m and $500$ m show to be in agreement with Kolmogorov even towards lower frequencies ($\approx 4 \cdot 10^{-3}$ Hz) as this is the case for the $w'$-spectra. This would imply, that turbulence is more

relevant for the local moisture variation at lower frequencies as it is for vertical winds. An atmospheric reasoning for such differences could be multifaceted. However, it seems plausible that external drivers such as e.g. solar radiation at the surface due to clouds in combination with horizontal advection can lead to a decoupling of the behavior of $q$ and $w$ in their accordance to Kolmogorov's law - especially over heterogeneous terrain.

The co-spectrum (Fig. 10) is the spectrum of the product $q'w'$. The latent heat flux is defined as $Q_E = \rho(p,T)L_v(T) * q'w'$,

where $\rho$ is the persisting pressure- and temperature-dependent air density and $L_v$ the latent heat of water vaporization. Thus, the co-spectra can be seen as test frequency spectra, giving answer to the question whether or not the instrument combination can ultimately lead to a proper determination of the latent heat flux. The actual calculation of those vertical fluxes, however, is not part of this publication.

Looking at the co-spectra, it becomes apparent, that the spectra of $q'$ and $w'$ do not always follow the same behavior as the

co-spectra can show different features compared to the single spectra. Especially in the altitude level of $500$ m, the co-spectrum consistently follows the Kolmogorov's -7/3-relation for higher frequencies ($> 2.5 \cdot 10^{-2}$ Hz). However, similar to the single spectra, also the co-spectra show frequency bands with almost constant energy levels. In contrast to the single spectra, the frequency bands in which deviations from the Kolmogorov law appear are now non-identical for the shown altitudes. From the analysis of more altitude levels, which are not included here, it could be observed that the intervals with strict accordance to the

dashed Kolmogorov line do change their frequency range non-systematically over height. Therefore, we conclude that those features are not the result of any systematic error. The co-spectra of all altitudes flatten towards lower frequencies, at latest around $< 5 \cdot 10^{-3}$ Hz, which is in agreement with a similar behavior that could be seen for low frequencies in both the DIAL humidity but mostly the Doppler wind turbulence spectra and, therefore, does not indicate a general DIAL deficiency at low frequencies. As a consequence to the observed flattening of the co-spectra, first of all, we conclude that the variations in vertical

latent heat flux slower than $5 \cdot 10^{-3}$ Hz are not generated by the decay of larger eddies, but represent the initial variability by the first order of large eddies or advected horizontal inhomogeneity. As the co-spectra do not show identical structures as within the initial spectra of $w'$ and $q'$ towards lower frequencies, we secondly conclude that the variations slower than $5 \cdot 10^{-3}$ Hz are independent and decoupled for $w'$ and $q'$ and have different origins. This hints towards the hypothesis that the variability of $q$, at least in high PBL altitudes, is to a large part a result of the horizontal advection of inhomogeneous air masses while the

variability of $w$ is dominated by convective dynamics. However, looking at the highest frequencies of the co-spectra, agreement with Kolmogorov's law is apparent at all altitudes, albeit minor differences in the frequency range of good agreement. This is

true even besides the general point that spatial averaging of the lidar data leads to losses of the high-frequency contributions of very small eddies (Brugger et al., 2016; Puccioni and Iungo, 2021). The advantage of co-spectra is, that the noise of $w'$ and $q'$ are not dependent of each other. This has the pleasing effect that higher noise levels from one instrument can be partly compensated by another instrument with better high-frequency quality.

As a result, the co-spectra show to be able to resolve turbulent structures down to the level of $5 \cdot 10^{-2}$ Hz or 20 s which is the Nyquist-Frequency for measurements with $\Delta t = 10$ s. Nevertheless, the unexpected behavior of almost constant energy levels over frequency remains to be of interest. As already indicated, in general, constant levels of energy in the co-spectra are interpreted as white noise and hint towards increased noise in the data - hindering the adequate detection of turbulence. Typically, and under the assumption that the highest frequencies are still within the inertial subrange, this white noise becomes apparent at the highest frequency range of the spectra where the measurements reach their SNR limit. Measurements with instruments at higher temporal resolution (e.g. Mauder et al., 2020a) show that the assumption of an inertial subrange is valid up to much higher frequencies than the herein presented maximum of $5 \cdot 10^{-2}$ Hz. For the herein presented case, however, both the single and the co-spectra match again well with the Kolmogorov co-spectrum for the highest frequencies in all altitudes. Therefore, we conclude that the observed "energy plateaus" are not necessarily due to insufficient instrument performance - which, however, can also not be ruled out categorically. Previous studies showed, that surface heterogeneity can lead to perturbations of the expected turbulent state (e.g. Sühring and Raasch, 2013; Kröniger et al., 2019). As a consequence, the imprint of surface heterogeneity on vertical fluxes is highly dependent on the structure of the surface heterogeneity itself (Kröniger et al., 2019; Mauder et al., 2007). Therefore, one possible explanation for the observed 'energy plateaus" in the spectra could be the imprint of surrounding heterogeneity on vertical fluxes of, especially, latent heat that are advected towards the lidar. The "blending height"-concept (Mahrt, 2000), describing a gradual, vertical decrease of influence from heterogeneous surfaces due to mixing and increasing eddy diameters, would be contradicting to this assumption. However, as shown by Sühring and Raasch (2013), influences by heterogeneous surfaces can indeed propagate throughout the entire PBL.

Translating the interval borders of the "energy plateaus" in the co-spectra from frequencies into time, the 'non-Kolmogorov' range reaches from $\approx 25$ - 33 s for the upper- and lowermost altitude ($3 - 4 \cdot 10^{-2}$ Hz), and $\approx 40$ - 66 s ($1.5 - 2.5 \cdot 10^{-2}$ Hz) for the plateau at lower frequencies at the upper altitudes. As can be seen from the single spectra within Fig. 9, those plateaus arise mainly from the spectra of $q'$ - although the spectra of $w'$ also show the previously described energy jumps, however, much less pronounced. Therefore, it can be assumed that there is much less advection of surface heterogeneity in the wind field. In the respective time period, wind data from the DWD site at MOL-RAO at $10$ m above ground shows a mean horizontal wind velocity of $\approx 3$ m/s at northerly wind directions from $\approx 340° - 20°$ (not shown here). Therefore, corresponding surface heterogeneities which could potentially cause the non-Kolmogorov behaviour, would be in the order of $\approx 75$ - 100 m and $\approx 120$ - 200 m. With a rough scale estimation for vertical wind speeds of $\approx 1$ m/s, such an input of surface heterogeneity would be in a horizontal distance of $\approx 1$ km (lowest altitude), $\approx 1.5$ km (middle altitude) and $\approx 3$ km (highest altitude). The naturally non-homogeneous horizontal and vertical wind field leads of course to alterations of this estimation. As the wind speed increases over height, the horizontal extent of surface-induced energy perturbations therefore most probably corresponds to larger horizontal inhomogeneities for higher altitudes. In order to evaluate the influence of the surrounding heterogeneity on

the co-spectra, Fig. 11 shows the surrounding surface conditions at MOL-RAO in Lindenberg. Sentinel-2 L2A satellite mosaic images from 20210715 - 20210725, with least cloud coverage as mosaicing order, are used to show the normalized difference moisture index (NDMI, Fig. 11a) and the RGB highlight optimized natural colors (Fig. 11c). NDMI delivers information on the water content in leaves and is therefore taken as a proxy for the evaporative potential. The RGB channel shows both the current state of vegetation and visually an estimation on the prevailing surface albedo. Fig. 11b shows the surface height in the region as taken from a image-based digital surface model (bDOM2022) by the local land surveying office LGB (Landesvermessung und Geobasisinformation Brandenburg) with a spatial resolution of $0.2\,\mathrm{m}$.

As can be seen, especially the NDMI map, which is probably the most relevant one in order to explain the energy plateau in the $q$-spectra, does show a large surface fragmentation in the region where the winds come from. For this measure, which does not provide information on the surface moisture itself, surface heterogeneities in the scale of $< 50\,\mathrm{m}$ to $> 2\,\mathrm{km}$ are indeed apparent. On an even larger scale (not shown by the maps), changing patterns of lakes and urban areas with horizontal extent $> 3\,\mathrm{km}$ are apparent within the wider surrounding of Lindenberg - potentially introducing non-typical energy contributions on even lower frequencies as well. A direct attribution of a specific surface pattern to the observed energy plateau is not possible as the wind field changes both in speed and in direction vertically. In order to better investigate, whether surrounding surface heterogeneity really is the causal link to the observed energy roll-offs in the turbulence spectra, more measurements at the same location have to be taken and investigated. If comparable behavior could be seen more often and, ideally and hypothetically by different sensors, the analysis of humidity turbulence spectra could offer a new approach to better understanding of PBL humidity transport processes. Nevertheless, keeping in mind that the co-spectra $w'q'$ can depend on several surface features, it seems plausible that the observed energy plateaus are not caused by white noise but by real atmospheric, turbulent transport processes. Therefore, we conclude that the measurement of vertical, turbulent fluxes of latent heat with the presented combination of instruments is most probably possible down to at least a frequency of $0.05\,\mathrm{Hz}$.

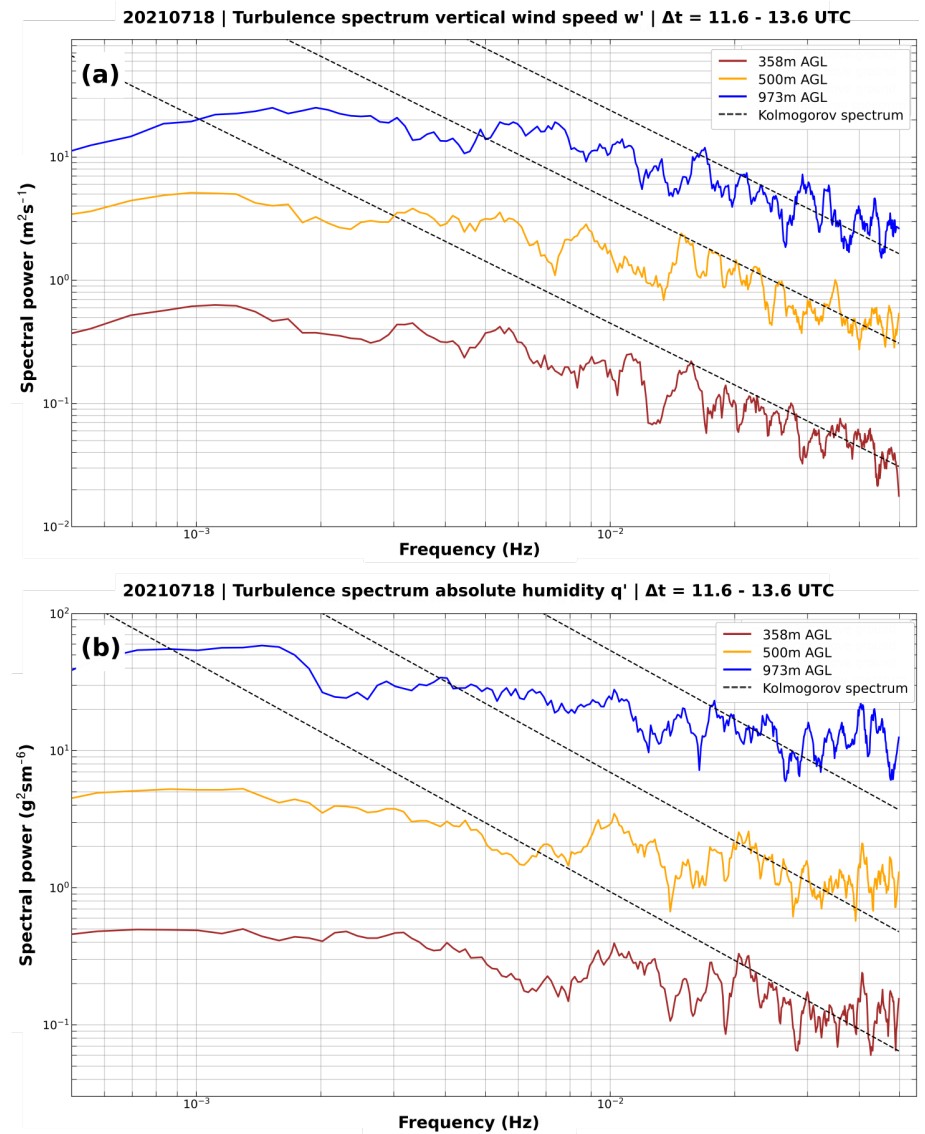

**Figure 9.** Turbulence spectra of (a) vertical wind $w'$ and (b) absolute humidity $q'$, calculated for the 2h time period 11.6 - 13.6 UTC in 3 identical altitudes which are all inside the PBL. For better visibility, the results have been vertically scaled. Also, the representation of the spectra is smoothed by applying a Savitzky-Golay filter of order 1 over a width of 10 bin. The dashed line shows the Kolmogorov -5/3-dependency.

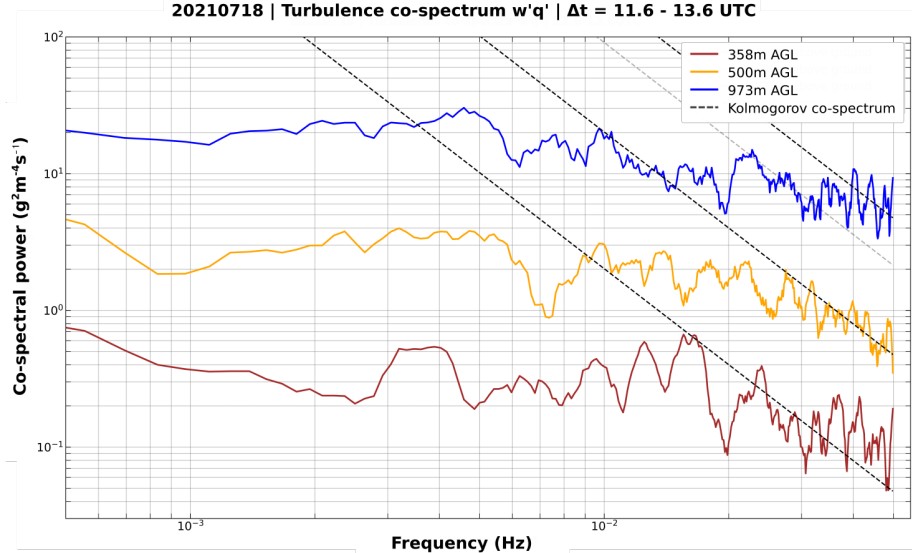

**Figure 10.** Turbulence co-spectra $w'q'$ with the same single spectra as shown in Fig. 9. The time period, altitudes, scaling factors and smoothing settings are identical to Fig. 9 as well. The dashed line shows the Kolmogorov -7/3-dependency for co-spectra.

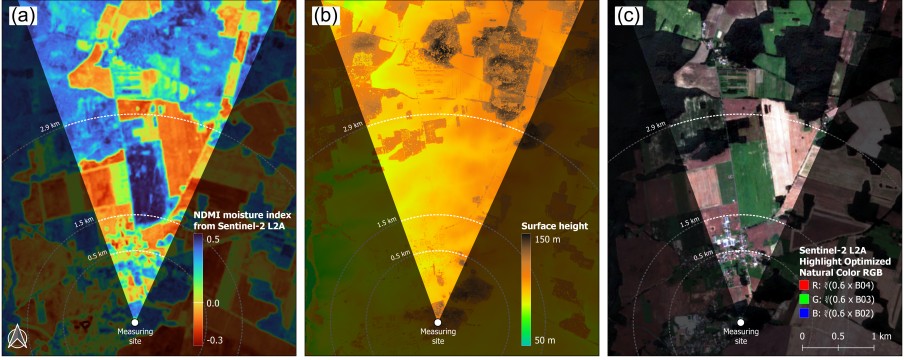

**Figure 11.** Overview of the surrounding landscape of MOL-RAO in Lindenberg. The highlighted circular segments correspond to the prevailing wind directions ($340\degree - 20\degree$) towards the ATMONSYS lidar in the circle center from 11.6-13.6 UTC on 18 July. 2021. (a) shows the Normalized Difference Moisture Index (NDMI) as measured by Sentinel-2, (b) the surface height over sea level from a digital surface model. (c) Shows the RGB channel, again observed by Sentinel-2. The Sentinel-2 mosaic data for (a) and (c) has been merged within the time span from 20210715 - 20210725 with the mosaicing order of least clouded pixels. This figure was kindly prepared by K. Winkler, KIT Campus-Alpin, IMK-IFU.

## 5 Instrument inter-comparisons

### 5.1 Inter-comparison ATMONSYS to RS41

Fig. 8 showed an intercomparison of the median ATMONSYS DIAL profile with a radiosonde ascent. However, the temporal overlap was not given as the radiosonde started at 10.75 UTC and the median was calculated for a time span of $\approx 70\,min$ and at a slightly later point in time. To achieve a more spatio-temporally representative intercomparison, free of any potential diurnal humidity development between the analyzed time intervals, we now show all single lidar profiles and their median for the same restricted time period of a bit more than 11 minutes. This period starts from the ascent time of the same radiosonde

until it reaches an altitude of $\approx 3.5\,km$ (Fig. 12).

Within this short and accurate time comparison, the profile-to-profile variance shows to be roughly of the same magnitude as it was in Fig. 8. Again, there are altitudes up to $3\,km$ in which the DIAL measurements show significantly lower variability than below. Therefore, we assume that the enhanced variability of humidity at $\approx 1.2 - 2\,km$ and $\approx 2.3 - 2.7\,km$ is real atmospheric variability which is captured by the DIAL and that is not related to a temporal trend over the previously shown time-span of

more than one hour. The relative vertical distribution of moisture between the DIAL median and the radiosonde data is very similar. However, despite the temporal overlap, the DIAL median mostly shows lower humidity concentrations as compared to the radiosonde. Only few profiles show as high humidity values as the sonde, whereas there are even altitudes (e.g. $\approx 1.2\,km$) in which not a single profile reached the humidity values by the radiosonde, not even within the statistical uncertainty of $4\,\%$ for the radiosonde measurements themselves. As Fig. 8 and the following Fig. 13 do not show a systematic issue of too low

humidity profiles, there is no obvious system-dependent explanation for this absolute deviation of humidity concentrations. Keeping in mind the substantial short-time fluctuations that can be seen within the DIAL data, as well as the measurement's low statistical uncertainty, those mismatches could principally be due to real temporal atmospheric variability. As a matter of fact, former studies have shown that humidity intercomparisons can be heavily dependent on both temporal and spatial overlap - even on very small scales (e.g. Vogelmann et al., 2011, 2015). For sonde-lidar intercomparisons, such spatiotemporal overlap

mismatches cannot be avoided. Therefore, the following subsection will show an extended intercomparison by the inclusion of ARTHUS and RAMSES lidar data during their overlap of operation in close vicinity to each other.

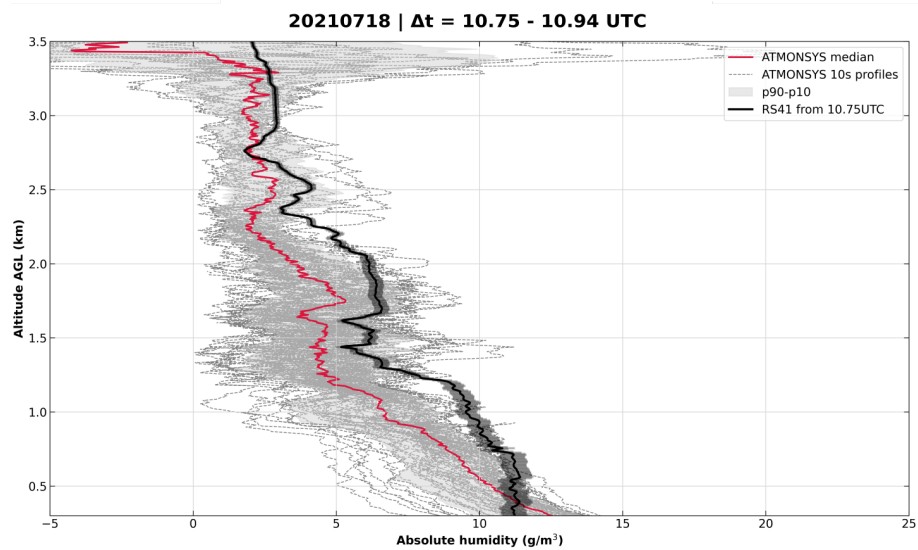

**Figure 12.** Inter-comparison between a radiosonde ascent and the ATMONSYS DIAL during the time of ascent from ground level to $\approx 3.5\,\text{km}$ above ground level. Statistical uncertainties for the radiosonde are represented by black shading.

## 5.2 Inter-comparison of ATMONSYS/ARTHUS/RAMSES to RS41

Lidar to radiosonde data comparisons, as shown in the previous subsection (and e.g. Ferrare et al., 1995; Reichardt et al., 2012; Späth et al., 2016; Muppa et al., 2016; Klanner et al., 2021), are a convenient method to evaluate the lidar data quality. Often,

it is the only way to test the lidar data against in-situ. However, even if the measurements of radiosonde and lidar show a good agreement, a certain degree of uncertainty remains. The spatial drifting of the radiosonde remains a relevant factor, and horizontal homogeneity cannot always be assumed from lidar data showing similar results to the radiosonde. Especially in the convective boundary layer during daytime, turbulent fluctuations cause significant sampling differences between the vertically pointing lidar instruments with diameters of the laser beams of a few centimeters and radiosondes which are drifting hori-

zontally while moving upwards with only a few meters per second. These instrumental sampling differences naturally cause differences in the measured data. So we cannot expect perfect agreement even if the uncertainties of each of the instruments were zero. Therefore, coinciding inter-comparisons between multiple lidars and radiosondes are of great advantage in order to better understand deviations between radiosondes and lidars. At least to our knowledge, such inter-comparisons have rarely or never been carried out with high-power boundary layer humidity lidars. Within the FESSTVaL campaign we had the opportu-

nity of such comparisons. In Fig. 13, we show an intercomparison between all three humidity lidars (ATMONSYS, RAMSES, ARTHUS) that have been operating in close proximity to each other at Lindenberg and radiosondes from the same location. Besides their proximity (Fig. 4), instrumental differences between the three lidars remain. One aspect is the difference in their power aperture products. This product of laser power and telescope area gives an idea how much signal one can expect from the different lidars, based on their optical setup (Tab. 4). Higher aperture products thereby potentially lead to higher signal

levels and better signal quality, if the background noise caused by larger telescopes can be reduced in an appropriate manner. However, it has to be acknowledged that the different measurement techniques of DIAL and Raman lead to differences in the magnitude of $\beta(r)$ (Eq. 1) and therefore a different instrumental behavior. Another difference between the instruments is both their way of calibration as well as the calibration data they're referring to. Whereas the DIAL data is free of any calibration, the Raman lidars have been calibrated against radiosonde measurements in Lindenberg. The RAMSES measurements use the

same calibration parameters for the entire second half of the respective month, therefore referring to a radiosonde from $\approx 2$ weeks earlier. The ARTHUS system, on the other hand, uses calibration parameters referring to the 00 UTC radiosonde on 18 July 2021. Fig. 13 shows two radiosonde ascents and the median values of all three lidars around the corresponding time

**Table 4.** Power aperture products of all three humidity lidar systems.

| Lidar | Output power (W) | Telescope diameter (m) | Power Aperture product (Wm$^2$) |
|---|---|---|---|
| ATMONSYS DIAL | 2 | 0.2 | 0.06 |
| RAMSES (near range) | 13.5 | 0.2 | 0.42 |
| RAMSES (far range) | 13.5 | 0.79 | 6.62 |
| ARTHUS | 20 | 0.4 | 2.51 |

of ascent. Note that the temporal resolution of the RAMSES lidar is 4 min, whereas the temporal resolution of the ATMON-SYS and ARTHUS is 10 s. Also, the vertical resolutions are different for all three lidars (Sec. 3). For the 18 UTC radiosonde

(starting time = 16.75 UTC, Fig. 13 a), the comparison time for all lidars has been shifted to slightly before the start of the radiosonde due to data gaps. Nevertheless, all 4 systems show good agreement in the lowest 2 km above ground. Besides too high DIAL values at the lowest altitudes, both the DIAL and ARTHUS values are in good agreement within their respective statistical uncertainties. The reason for the steep humidity increase in the DIAL data towards ground might be detector issues which will be discussed later within this subsection. The RAMSES profile reveals smoother data due to longer integration

times and, at the same time, highest humidity values from $\approx 0.6$ - 1.5 km. Whereas the agreement between the radiosonde data and all 3 lidars is quite good at $\approx 0.6$ - 1.1 km, there is also an altitude range ($\approx 1.3$ - 1.7 km) in which the radiosonde shows substantially lower values than the 3 lidar systems. At $\approx 1.7$ km, a steep gradient in humidity can be observed both within the radiosonde and the lidar data - with some minor differences. ATMONSYS and RAMSES measure the gradient with almost identical steepness. ARTHUS, on the other hand, also shows the gradient, however, at higher altitudes and consistent with the

radiosonde. For the radiosonde, possible explanations for increased humidity measurements towards higher altitudes could be hysteresis effects stemming from a wet balloon surface or spatial mismatch. This, however, does not explain the differences between the ARTHUS data and the data measured by RAMSES and DIAL.

    Overall, starting from $\approx 1.8$km, it can be seen that during daylight conditions, ARTHUS measurements show a substantial increase in noise and statistical uncertainties. This behavior can be attributed to the physical constraints of the Raman technique

during daylight conditions at this high sampling frequency. Above an altitude of $\approx 2.7$km, ATMONSYS measurements show

lower humidity concentrations which are not seen by both RAMSES and the radiosonde, indicating the maximum range for the DIAL measurements under the given conditions. Beyond this altitude, only the RAMSES data shows realistic humidity values in good agreement with the radiosonde.

The second intercomparison is chosen for the midnight radiosonde ascent (starting time = 22.79 UTC, Fig. 13 b). As already discussed in Sec. 3, an aerosol layer of different origin and dense clouds were present during that night and also directly at the time of the radiosonde ascent. In order to capture more profiles without any cloud influence, the lidar comparison time has been shifted to a time period of almost 40 min later than the radiosonde starting time. Especially during night, the lack of convection generally leads to more continuous humidity structures, justifying this temporal shift. On first sight, a very good agreement

between RAMSES and the radiosonde can be observed over the entire range of altitude, with measurement differences mostly within the respective statistical uncertainty. Interestingly, both ATMONSYS and ARTHUS see lower humidity concentrations than RAMSES and the radiosonde at lower altitudes, outside the statistical uncertainties. In comparison, the DIAL data shows the lowest humidity values, including a small humidity minimum at 1.6 km due to unfiltered wisps of clouds. In contrast to Fig. 13a, the issue of unrealistically increasing humidity concentrations at the lowest altitudes is not that much apparent at this

point of time, indicating problems with nonlinear signal behavior mostly during daylight conditions and thin clouds. In case of optically dense clouds, causing a detector overload, the corresponding data is flagged by the transient digitizer within the ATMONSYS data processing routine. Such clip flags have not been set by the transient digitizer in the corresponding time. However, the PMT detectors in photon counting mode already show non-linear behavior before full saturation (dead time issue). But, also in analogue mode, the preamplifiers of the transient digitizer can show a non-linear behavior already at 50% of

the clipping threshold caused by atmospheric fluctuations in single shots (B. Mielke, Licel, priv. comm.). A closer look at the DIAL raw data (which is not shown within this publication) revealed non-linear looking structures in some signals. Therefore, we conclude that the false maxima of humidity concentration in this special case as well as the overestimation of humidity in the lowest altitudes within Fig. 13a are due to signal levels that haven't been high enough to entirely saturate the detector but already led to signal distortions. However, at least in comparison to the humidity values of the radiosonde and ARTHUS, there

is still a more pronounced increase in the ATMONSYS humidity towards the lowest altitudes within Fig. 13b. Therefore, it seems plausible that this feature is not necessarily caused only by daylight background but could also be partly caused by stray light from the outgoing laser pulse. Apart from this deficiency, all three lidars, and also the radiosonde, prove to be very consistent to each other at altitudes $\approx 2$ - $2.5$ km. This good agreement and the fact that measurements below those altitudes have been in much better agreement 7 h earlier justifies the assumption that those deviations partly really are a matter of different

temporal averaging, and therefore atmospheric variability, rather than systematic nature. This calls for the analysis of a larger intercomparison dataset, which, unfortunately, was not possible to collect at that point of time.

Despite a good agreement between the sensors at changing altitudes and times, deviations between the four measurements of $\approx$1-3 g/m$^3$ at all heights, corresponding to relative deviations of $\approx$10 % and more, remain. In the case of the DIAL measurements, statistical uncertainties have shown to be lower - hinting towards atmospheric variability on both small temporal and

spatial scales. This, in conclusion, highlights the valuable additional information which can be provided by humidity lidars at

high spatio-temporal resolutions. Up to now, only very sparse information on the extent of short-term humidity fluctuations in the middle and upper PBL exist. Future humidity measurements at 10 s resolution and less, such as introduced here for the ATMONSYS DIAL, will therefore deepen our understanding of turbulent exchange processes and the energy budget throughout the PBL.

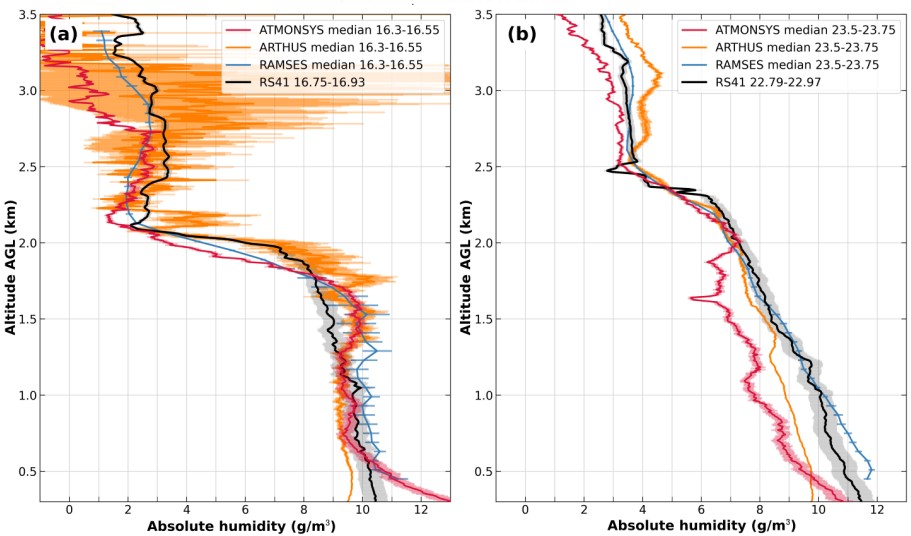

**Figure 13.** An intercomparison of all three humidity lidars (ATMONSYS: red line, RAMSES: blue line, ARTHUS: yellow line) and the radiosonde (black line). Shown are median values from a time intervall of 15 min around the afternoon ascent (a) and the midnight ascent (b), referencing to UTC. The temporal resolution of ATMONSYS and ARTHUS is 10 s, the data of RAMSES is shown with a resolution of 240 s. The radiosonde is at a temporal resolution of 1 s. The statistical uncertainties of the data are indicated by shading.

## 6 Conclusions

For the first time, we presented data from the mobile high-power ATMONSYS humidity DIAL, incorporating a novel Ti:Sa laser concept. The ATMONSYS DIAL system is designed for boundary layer humidity measurements with the goal of resolving turbulence at a sampling frequency of 10 s and vertical resolutions of less than 200 m at 3 km above ground. In this manuscript, we demonstrated that the system is capable of stable operation, reaching far beyond the planetary boundary layer top under clear sky conditions, even during daytime. The presented data shows that a significant concentration of aerosols, resulting in stronger backscatter than pure Rayleigh backscatter from air molecules, allowed for a maximum measurement range of $\approx 3.2$ km above ground around high noon in summer without clouds. During night, the maximum range is slightly increased as Rayleigh backscatter alone yields a sufficient SNR without daylight background. The instrumental uncertainties have been demonstrated to stay below a level of 5 g/kg. A spectrum analysis of the DIAL showed good agreement with the spectrum analysis of the Doppler wind lidar. Both spectra are following Kolmogorov's "-5/3 law" for a broad frequency range, however, with some portions where the energies drop in their absolute level before they again follow the Kolmogorov law. However, espe-

cially at the maximum frequencies of $5 \cdot 10^{-2}$ Hz agreement with Kolomogorov's law has been proven. Therefore, we conclude that interruptions to the Kolmogorov behavior could be caused by perturbations due to the very heterogeneous surrounding of the measurement site. With the limited amount of data that has been collected by the ATMONSYS during this particular measurement campaign, a deeper causal investigation has not been possible. If, however, similar behavior could be reproduced by future measurements, examining the deviations from Kolmogorov behavior in turbulence spectra could open new ways of transport process analysis. The combined turbulence analysis of humidity data with vertical wind data from a Doppler wind lidar, which is the important measure for vertical fluxes of latent heat, proved that the two combined systems are capable of resolving turbulence at the sampling frequency of $10\,$s. This offers new possibilities for measurement campaigns and involved modeling based on large eddy simulations. For lower frequencies at $\approx \cdot 10^{-2}$ Hz, wind and humidity show different spectral behavior which does not propagate into the co-spectra. From this we conclude, that those two measures behave independently at low frequencies and that their variability has different sources that are not predominately driven by eddy decay. Thus, under the given convective conditions, we assume that vertical wind variability at lower frequencies is more a result of convection while the variability of humidity is more dominated by the advection of heterogeneous air masses. The inter-comparison of absolute humidity values from the ATMONSYS DIAL with accompanying humidity lidars and radiosondes showed overall good agreement. However, a potential problem with non-linearities, perhaps caused in part by overload of the transient digitzer pre-amplifiers, has been recognized under certain conditions as e.g. fragmented wisps of clouds passing over the lidar. This, together with steep gradients of aerosol concentration, has shown to be potentially problematic for DIAL humidity measurements, especially in cases as the herein presented where a complete implementation of a full Rayleigh-Doppler correction is not included as it is not straightforward. Frequent and non-systematic deviations between the median values of all lidars and the radiosonde of 1-3 $\mathrm{g/m^3}$ during a time span of $15\,$min hint towards considerable short-term humidity fluctuations beyond the statistical uncertainties of the instruments. We conclude that humidity lidar systems, such as the presented ATMONSYS DIAL, provide the opportunity for more accurate information on the locally prevailing atmospheric state and its short-term variability.

*Data availability.* The data used in this publication is available upon request.

*Author contributions.* JS and HV were involved in the development of the ATMONSYS system, its operation and the scientific data analysis. JR gave decisive comments that shaped the scientific quality of the data analysis. MM introduced important methodological concepts for the manuscript. JR, KW and AB operated their instruments and provided their processed data. All authors were involved in the writing process of the manuscript.

*Competing interests.* The authors declare that they have no conflict of interest.

*Acknowledgements.* First, we want to thank Dr. Frank Beyrich from DWD in Lindenberg for both his big support and organizing talent during the FESSTVaL campaign as well as his experienced review during the scientific refinement of this manuscript. Furthermore, we thank Matthias Perfahl for his enduring technical support and proficiency both during the development of the ATMONSYS system and its operation in the field. Finally, we thank Dr. Karina Winkler for her valuable input regarding both the data visualization of Fig. 11 and general wording.

Also we thank for the financial support of this work. This work has been partly funded by the German Federal Ministry for Digital and Transport (BMDV) within the framework of the DWD program for extramural research, project: Quality Assessment of ground-based Lidar measurements in the Boundary layer (QALiBo): evaluation and verification of scanning strategies, quality tests and uncertainty quantification / contract number: 4819EMF05. Furthermore, this work was funded by the Federal Ministry of Education and Research of Germany within

785 the ACTRIS-D project (grant no. 01LK2001B) and by the Helmholtz Changing Earth – Sustaining our Future research program within the Earth and Environment research field and finally by the Deutsche Forschungsgemeinschaft (DFG, 406980118, VO2423/1-1).

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
