# Peer review of "The ATMONSYS water vapor DIAL: Advanced measurements of short-term variability in the planetary boundary layer"

_Atmospheric Measurement Techniques, 2024_

## Author Comment (AC2)

**Review comments #2:**

**General:**

Many thanks for the paper "The ATMONSYS water vapor DIAL: Advanced measurements of short-term variability in the planetary boundary layer".

This paper consists of two main ideas:

a) the description of the system ATMONSYS water vapor DIAL and

b) "claiming" of resolving turbulent latent energy fluxes (in different sections of the paper).

I would withdraw the paper and would rewrite it containing part a) as

part a) needs more information and

part b) is not sufficient. The unit of the turbulent latent energy flux is W/m2. There are no (height-dependent) latent energy flux values presented in the paper.

I would need some more explanation regarding the topic "the paper present novel concepts, ideas, tools, or data".

There are already published water vapor DIAL systems and (even in more detail) latent energy flux measurements with DIAL systems.

Please, check

Christoph Senff, Jens Bösenberg, and Gerhard Peters, Measurement of Water Vapor Flux Profiles in the Convective Boundary Layer with Lidar and Radar-RASS,

https://doi.org/10.1175/1520-0426(1994)011<0085:MOWVFP>2.0.CO;2

Behrendt, A., Wulfmeyer, V., Senff, C., Muppa, S. K., Späth, F., Lange, D., Kalthoff, N., and Wieser, A.: Observation of sensible and latent heat flux profiles with lidar, Atmos. Meas. Tech., 13, 3221–3233, https://doi.org/10.5194/amt-13-3221-2020, 2020.

Many thanks for this review. The comments helped to significantly improve this manuscript. We carefully revised the manuscript based on the comments by both reviews. When going through the individual points of criticism, we came across a small but relevant error in the previous calculation of water vapor absorption coefficients. Therefore, all plots including DIAL data were recalculated and are now in their updated version. As a result, some of the comments do not apply anymore.
A detailed explanation on the changes made, due to the recommendations and requests, is following below.

**In more detail:**

- Lines 40-42: It seems that a permanently measuring system for the WMO is the goal of the ATMONSYS developers? Otherwise it makes no sense to refer to the requirements of the WMO.
    - The reference to the WMO requirements seems to have been misleading. As the ATMONSYS lidar is an experimental development, we don't claim it to be an operational instrument that immediately helps in the improvement of numerical weather prediction. However, from a developers perspective, the WMO requirements can be seen as goal system performance parameters. Recent studies incorporating single high-resolution water vapor lidars show persisting effort in the improvement of model parameters based on enhanced knowledge gained by lidar systems. Therefore, we would argue that the WMO observational requirements are a fair motivation. A respective sentence, clarifying that the ATMONSY lidar isn't thought to be operational, has been added (l. 42-44/736-737).

- Only absolute values of the required uncertainties are given at the web page of the WMO. But, the authors indicate 5% which is a relative uncertainty. The truck in Fig. 1 looks like a mobile system which could also be used at different areas. In more dry areas one would need a system with an uncertainty of less than 2-10 g kg-1 (marked/breakthrough, threshold values of the WMO for altitudes at near surface). Aside, I didn't find requirements defined for air specific humidity at larger altitudes at this web page. I know that in arid regions the water vapor content can easily be below 1 g kg-1 (at ground and at lofted altitudes). Given the presented lidar ATMONSYS, I don't understand the argument of line 40-42.
    - We apologize for a mix-up of numbers due to different sources. Indeed, the given uncertainty numbers are given as absolute values. The numbers for both uncertainty and temporal resolution have been adapted to the actual entries for "high resolution numerical weather prediction inside the PBL" as they are noted on the respective website. As we show in Sec.4.2, the statistical uncertainty of our measurements is below 5%. Therefore, regarding the statistical uncertainty, the ATMONSYS uncertainties should even fulfill the WMO "goal", regardless of arid or moist regimes, which is due to the measurement principle itself. A respective remark has been added to l. 238-239. References to the WMO values are changed within the entire manuscript.

Sec. 2.2:

- Why is no beam expander installed? Is the emitted light polarized? If yes, how is the transmission of the different polarization directions and the retardation (of all used channels) in the receiver?
    - Good point. This idea was discussed during the setup but ultimately dropped. An explanation is added in l. 132-135.
    - The missing information on the polarization of all wavelengths was added in l. 131).
    - The retardation of the s-polarized Raman signal is now included (l. 145-148).

Sec. 2.3:

- Line 158: The digitization range of the transient digitizer 12 bit, yielding to the dynamic range of the DAQ of 1/4096. The detector coupling to the digitizer needs to be explained (compare line 640). The AOD-dependent maximum range of water vapor measurements (lambda_on) should be given with this number and a standard atmosphere at mid-latitudes.
  - This part of the description was probably misleading. Although the digitizer works with 12 bit, the memory depth of the digitizer is 24 bit, from which one bit is taken as flag clip in case of signal overflow. As we operate the system only for 10s (1000 shots), we effectively only use a maximum memory depth of 21 bit. Therefore, the effective resolution of 1 bit would be 500 mV/$2^{21}$ ~ 0.2m µV, which is below the transient digitizer noise of ~0.5mV. We hope this explanation, together with the changes in l. 164-168 meets your concerns.
- Lines 189-190: How stable are the emitted wavelengths? How is the seeding working? Which control is installed in the Ti-Sa that it follows DL1 and DL2 from shot to shot? How is the wavelength of the emitted light monitored?
  - At this point a reference to the previously cited paper Vogelmann et al. 2022, including detailed information on those features, is added.
- Lines 208-210: The "logistic function" may be necessary, but needs more explanation. There must be a "physical motivation"! Otherwise, the authors should avoid discussing data below 1.1 km altitude. As I read, the median value is fitted to the "closest available radiosonde". Hence, it seems to be a height-dependent calibration? What is done in case of a variable water vapor profile (known and also later shown). What is done if radiosonde data are not available?
  - With the newly calculated data (not being affected by erroneous absorption coefficients anymore), the DIAL data behavior changed slightly. Nevertheless, during some (but not all) measurement periods, data still shows odd behavior at the very lowest altitudes. We still assume that this is an issue of some detector overflow but, considering your justified remarks, we decided to not do any corrections on the data anymore.
- At detailed error analysis of the ATMONSYS data is missed! This is an absolute prerequisite for a newly constructed system.
  - We agree. So far, there has been a error analysis included within Sec. 4.2. However, it seems that this Section wasn't explained in a sufficient manner. Therefore, we now included a more detailed explanation of our error calculation within Section 2.3.

- Table 2: More information is needed to the errors, the measuring range (under certain conditions). The authors didn't show data until 4.5 km altitude. That's why, I'm a bit anxious about this value in the table. Beam divergence of the laser: please, indicate that the given value is the full angle beam divergence (some scientists think about a half angle when using the words "laser beam divergence").
  - Information on the statistical uncertainty and measurement range are now given. The values are a conservative estimation based on realistic simulations. Changes in SNR, (distribution of) humidity, (distribution of)

aerosol, and choice of the actually tunable wavelength for $\lambda_{on}$ , however, do alter the specifications.

- For consistency reasons with the shown data, both Table 2 and Fig. 3 now only refer to a max. altitude of 3.5km
- The divergence is describing the full angle, which is now clarified within the table

Sec. 3:

- Lines 230-244, RAMSES lidar: I didn't find the measurement errors and the vertical resolution in the description.
    - A respective sentence has been added at the end of the "RAMSES"-subsection.
- Lines 245-252, ARTHUS lidar: I didn't find the measurement errors in the description.
    - Details on the error calculation, including respective references, have been added to the "ARTHUS" subsection.
- Lines 253-265, Doppler wind lidar: I didn't find the measurement errors in the description.
    - A respective sentence has been added at the end of the "Doppler wind lidar"-subsection.
- Lines 259-260: I didn't get the content of the sentence "We removed the data with a high noise level by filtering with a relatively low Signal-to-Noise Ratio (SNR) + 1 threshold of 1.000 to keep the data availability high."
    - This admittedly strange value, still leading to filtering of data, comes due to the wind lidar's system-internal calculation of SNR values. A short sentence of explanation has been added to this line.
- Lines 266-270, Radiosonde: I didn't find the measurement errors in the description.
    - Information is now added.
- Lines 271 and follows: Measurement day: 18 July 2021.
    - All occurrences are changed accordingly.
- It seems to me that a new air mass has moved over the old one during this day as the depolarization of the particles (lines 284-286) and the wave structure in Fig. 7b indicate. Hence, it might have not been the best day for presenting the data.
    - We agree that there is a change in aerosol type at higher altitudes during the day, which could also indicate a change of air masses at those levels. However, the air mass inside the PBL seems to stay the same during the day, which is the time range where we investigated general system capacities, short-term variability and turbulence characteristics.
      For the intercomparison between the different lidars and sondes (Fig. 13), to our opinion, it should not be disturbing whether or not there is a change in air masses between the two shown timesteps.

Fig. 5:

- There is a need to specify the local time relation to UTC since the local time indicates when the sun is at its highest and one may expect the largest height of the PBL. It seems that the largest height of the PBL is at 16 UTC which equals to 18 local time. Please, correct me and explain the details

- The information on local solar noon (~11 UTC) has been added to the caption.
- Regarding your second point, in accordance to comparable criticism by Review#1:
  The PBL height determination purely based on the gradients of aerosol/humidity may lead to deviations in comparison to the thermodynamic/kinetic energy PBL height.
  To our understanding, a proper PBL height determination based on wind data requires horizontal wind information to identify the low level jet nose. However, the Doppler wind lidar next to the ATMONSYS system has been operated only at vertical stare. Calculating the temporal standard deviation of the vertical wind on each height leads to vertical profiles of $\sigma_w$. However, it seems to be the case that an automated PBL height determination based on this measure very much relies on personal choices for thresholds (e.g. : https://doi. org/10.16993/tellusb.1876).
  As a work around, we performed PBL height calculations based on the bulk Richardson number with the available radiosonde data. The respective values are included into the manuscript (Tab. 3).
  Above that, there was a VAD scanning Doppler lidar approx. 7 km away producing horizontal wind speed data. The sparse data that is available from those measurements, however, confirms a maximum in wind speed at around 1500m agl during that time.
  Therefore, we trust the radiosonde data as a rough estimation of the PBL height development.
  A table with the calculated values as well as explaining sentences have been added to the subsection "Measurement day: 18 Jul 2021".

- It would also be possible and helpful to indicate the PBL height in one of the charts.
  - We decided to include the non-continuous bulk Richardson number PBL heights in Table 3. Explanations to this decision are given in l.332-341.
- The rainbow scale of the colors in charts b and c is not suited for the given values (especially not in the PBL).
  - We're not quite sure how to interpret this comment. In both cases b and c, the scale covers the entire value range.
    Especially with respect to Fig 5c: In our opinion, it should not be of utmost importance to distinguish between one-digit values, but rather to recognize the apparent change in aerosol types, which is enabled by the chosen scale of colors.
- Why do some charts show clouds and others not as all charts are from one system?
  - The clouds are prominent in the particle backscatter coefficient, but only vaguely discernable in the particle depolarization ratio because in water clouds, depolarization is close to the molecular background. Water vapor mixing ratio and fluorescence backscatter coefficient are not directly related to clouds and thus show no cloud signature.
- I missed a chart of the ATMONSYS data in this Fig.!
  - We'd also like to have one. But regardless of our own wishes,  ATMONSYS did not measure uninterrupted during that day.
    Therefore, for the sake of good visualization, we only use the uninterrupted

> RAMSES data to deliver a general overview of the atmospheric conditions of the measurement day and use the ATMONSYS data only for specific time intervals.

- There might be differences between the values measured by ATMONSYS and RAMSES (comparing Figs. 5a and 6a; time between 12 and 12.6 UTC; at least in the altitude range between 2.5 and 3.5 km)?
  - There are two aspects that have been changed:
    - First, the former units for humidity in Fig. 5 have been g/kg – in contrast to g/m³, which is used for all the other plots. This has been changed to guarantee consistency
    - Second, as explained earlier, the former DIAL data has been afflicted by falsely calculated sigma values. This has been changed. To meet the criticism regarding a correction of values in the lowest altitudes, the DIAL values – in this raw representation – are too high below 0.5 km at this point of time.
      At higher altitudes, values are of similar magnitude.

Sec. 4:

- Lines 306-308: Never trust colored plots when looking for data out of the colors. But its okay, PBL top is ca. 1.2 km.
  - You're right. Those lines have been changed in accordance to the radiosonde based PBL top calculations  (see remarks for Fig. 5).
- Lines 317-319: The same explanation is necessary for the wind.
  - Thank you for pointing this out. The wind representation is in absolute values – a corresponding explanation has been added.
- Line 322: delete "drastically" as it is "only" the factor of 2.
  - Deleted.
- Lines 325-327: The wind direction changed at the PBL top. Could this indicate a different air mass?
  - There might be a misunderstanding. The presented data only refer to vertical wind speed, not to horizontal wind directions. The Doppler Lidar next to the ATMONSYS lidar was in vertical-stare only and, therefore, cannot provide horizontal wind information.
    The radiosonde wind data does not show a particular change in horizontal wind directions apart from a regular Ekman spiral behavior.

- Lines 350-353: What a pity.
  - Indeed...

Fig. 6:

- Could you please confirm that the data are larger than the system noise level?
  - Yes.
    This can also be seen from Fig. 8 (right side, blue line), where the median value of all statistical single-profile relative standard deviations from the 1h

period is plotted. An explanation on how this statistical error is calculated, meeting your above request on a more detailed description of uncertainties, has been added to Sec. 2.3.

- Could the "wave-like" structure of the particle backscatter coefficient between ca. 1.8 and 2.5 km altitude be caused by the slightly existing wind sheering?
    - Fair question, which we cannot answer for sure, as is now included in the manuscript (l. 365/366)
    Possible explanations could be wind sheer, orographic obstacles leading to waves or maybe even convection-driven vertical momentum that is propagating into higher altitudes.

Fig. 7:

- a) looks much noisier than the other charts especially > 1.2 km. Why should the water vapor show different structures? Would it be possible to smooth the water vapor data in altitudes > 1.2 km?
    - Good point. First, there has been a small mistake in the caption of Fig. 7. Originally, it stated that Figs. a/b show relative deviations (in contrast to the scale description of a, which is actually right). As a matter of fact, Fig. 7a really shows absolute deviations. The caption text has been corrected accordingly. The reason for absolute deviations in Fig. 7a is to enable a visual pattern recognition within the PBL. The colors there would be much more transparent if the visualization would be changed to relative deviations as well.

    To our understanding it is plausible that the fluctuations of water vapor and aerosol are independent from each other as they have different sources and, more important, different vertical distributions. As can be seen from Fig. 6b, Aerosol concentrations reach comparable values above 2.5 km as below 1 km. Humidity, in contrast, shows a general decrease over height. Therefore, water vapor looks much noisier above the PBL.

    In principal, it would be possible to apply a data smoothening above 1.2 km. We attached the corresponding plot, where, above 1.2km a temporal, vertically linear increasing, smoothening is applied. The increase of the smoothing interval starts with 10s at 1.2km up to 90s at 3km altitude. However, we fear that such a representation could lead to misunderstandings in its interpretation as it would be inconsistent to the other subplots.

[Figure]

Sec. 4.2:

- This section is not straightforward. There are no formulas given at all. For me it would be helpful to have first insights into the errors of the measurements (together with an error propagation analysis) and afterwards a second subsection with observations of the atmospheric variability.
    - Thank you for this feedback. We agree that the introduction of error propagation was too sparse and that it would make more sense to introduce them at an earlier point within the manuscript. Therefore, we included a detailed explanation on the calculation of the statistical uncertainties in Sec. 2.3. – now also represented by its title. By this, Figs. 7/8 should be much better understandable.
- Fig. 8 and all the explanations regarding this Fig. make no sense with the sentence at lines 425-427.
    - Allow us to convince you otherwise. Only shortly after the radiosonde ascent, for which the data is shown within Fig. 12, the DIAL data is afflicted by clouds. Therefore, another consistent  time period had to be chosen for the data analysis covering a longer time period. The time period 11.6-12.6 has been chosen consistently for most of the other plots as this was a time period without major cloud occurrences nor any major weather change. Including the radiosonde ascent data from 1 h earlier is done for two reasons: (1) to get a general idea whether or not the humidity structure changed entirely - which is not the case.
    (2), Fig. 8 shows a profile-to-profile variability within the DIAL measurements. The idea of this representation is to get an impression to what extent this variability can be explained by instrumental noise or "atmospheric noise" - which is then discussed for this Figure (e.g. based on the "narrowing" of variability at higher altitudes).

- Line 423: Usually, the ascent speed of a radiosonde is 5 m/s. This means, that the radiosonde reaches an altitude of 5x60x5 m = 1500 m in 5 min (compare also line 587).
    - Thank you for spotting the mix-up of numbers. Up to the altitude of 3km – where noise starts to become problematic – the radiosonde needs 10 min.

Sec. 4.3, Turbulent spectra:

- Lines 430-431: There is not only the mechanically caused turbulence but also the thermally.
    - Indeed. This information has been added.
- Line 432, "strong winds in the free troposphere": the wind speed is almost the same in the upper PBL and the lower free troposphere (< 2.5 km)?
    - "Strong winds" has been changed to "Stronger winds". The horizontal wind speed from that particular day isn't visualized within this paper. However, horizontal wind speed generally increases towards higher altitudes due to lower friction, typically reaching a local maximum at the "low level jet". This behavior is validated by the respective radiosonde data which is not presented in this manuscript.
- Lines 432-433: Please, numbers of eddies diameters and PBL heights => for crosschecking.
    - Added.
- Lines 438-439: Please, refer to the two given references in more detail.
    - Thank you for reminding us of the Senff et al. 1994 publication, which has been gotten off our memory. We included it in here as well and made the sentence more specific
- Line 448: How long did you average? With other words, how long did you assume a "frozen atmosphere" (Taylor hypothesis). The same equation needs to be defined for the quantity "wind vertical velocity".
    - The information of 2h has been added in the subsequent sentence, the equation for wind is now defined as well.
- Line 463: "good agreement" => how good?
    - The statement "good agreement" is based on  visual perception, which is now clarified for the respective sentence.
- Lines 474-475: The selected measurement day seems to be not optimal? Why not taking another day?
    - Following this logic, such analyses would exclusively make sense for static atmospheric states over absolute homogeneous terrain, in the end targeting for the smoothest turbulence spectra in "lab-like" environments.
    From personal conversation with other humidity lidar operators, it is quite common that perfect Kolmogorov behavior isn't achieved. We reason, and discuss this within the manuscript (Sec. 4.3), that it is worth looking at non-ideal cases and investigate whether such analyses give information on turbulence chracateristics over heterogeneous terrain – potentially delivering a good method to better understand transport processes if future measurements would show equal behavior. To our knowledege, this is a novel perception to the topic of Kolmogorov anlaysis for humidity lidars, calling for further, future, investigations.
- Line 480: "Thus, the co-spectra can be seen as frequency spectra of the latent heat flux." => but the latent heat flux is more than spectra. It is a height-dependent value in W/m2. The error analysis (including error propagation) is missed.
    - The formulation has been misleading. The respective sentences have been changed and extended (l. 551-555).

- Lines 506-507: The measurement equipment is only useful if its noise is less the "noise" of the observations. I feel, that the white noise from the instruments in the frequency spectra must be observable at higher frequencies and/that the -5/3 dependency is resolvable at lower frequencies than the white noise.
    - We agree. This sentence is formulated in a misleading way. From a pure theoretical perspective, assuming that the temporal resolution of the instrument is much higher than 10s, one would leave the inertial subrange with the -5/3 dependency towards high frequencies. This would ultimately lead to white noise behavior.
    However, as can be seen by turbulence analysis for wind measurements (.e.g. in https://doi.org/10.5194/amt-13-969-2020), the inertial subrange reaches much deeper into high frequencies than can be shown with data sampling every 10s.
    However, there could already exist white noise at the shown frequencies if the instrumental noise would be too high. To our understanding, this isn't the case as the -5/3 behavior can be seen at the highest frequencies that are resolved by the 10s measurements, validating the ATMONSYS DIAL data quality for turbulence investigations at the given resolutions.
    The sentence has been slightly reformulated to make our point clearer.

Fig. 10:

- The results are not understandable. Why is the co-spectral power largest at the lowest frequencies (largest eddies) at 358 m AGL?
    - Please note that these are co-spectra that are not pre-multiplied with the frequency; therefore, a drop-off at low frequencies is not necessarily to be expected, cf. Fig. 8.9(e) of Stull (1988).

[Figure]

    - This would be different for pre-multiplied co-spectra, cf. Fig. 8.9(f)

[Figure]

- Why differ the results so much from the observations of Senff et al. (from 10 July 1991, PBL top at 1100 m)? Both observations (Senff et al. and Speidel et al.) are made during summer time and the boundary layer heights are almost the same.
  - Regarding the co-spectra, the explanation is given within the previous point. Nevertheless, looking at the general structures, to us, the results aren't necessarily that different, but a comparison might be complicated by differences in the resolved frequencies. Senff et al. measured with a temporal resolution of 60s and a vertical resolution of 75m. It might well be, that those resolutions weren't sufficient to resolve a Kolmogorov-behavior within the co-spectra. Looking at Fig. 10 from Speidel et al., obvious Kolmogorov-like behavior only starts at frequencies $>10^{-2}$ Hz. This part of the spectrum wasn't resolved by the 60s measurements from Senff et al.
  - In addition, taking into account the reasoning that surface heterogeneity can lead to different spectral characteristics, measurements at different locations might as well lead to general deviations.
    In the specific case of the local measuring site in Lindenberg, there are surrounding landscape structures that could even more explain your previous point. Considering that the lowest shown frequencies in Speidel et al. belong to times of a bit more than 0.5 h (1000-2000s), and taking into the account the horizontal wind speed of 3m/s, this would correspond to structures of ~3-6km. Both the "Scharmützelsee" (Lake Scharmützel) as well as the town "Fürstenwalde/Spree" are in the direction from where winds on that day were coming from. It doesn't appear to be unrealistic, that such structures cause low-frequency contributions to the spectra. (Sentence added to l. 619-621.

- The section is titled "Turbulent spectra". This represents the presented results.
  - Ok.
- The unit of the turbulent latent energy flux is W/m2. There are no (height-dependent) latent energy flux values presented in the paper.
  - True. The scope of this paper is the introduction of the ATMONSYS lidar for the observation of humidity short-term variability. In order to categorize its general suitability for the calculation of latent energy fluxes we performed a general turbulence spectra analysis. The atmospheric interpretation of calculated latent energy fluxes, to our impression, would be interesting in another publication with pure focus on atmospheric processes. This clarification has been added to (l. 551-555).

Sec. 5

- I would like to see more detailed discussed inter-comparisons with other measurements to proof the performance of the ATMONSYS.

  So do we, however, there wasn't a larger temporal overlap between the three different instruments.

Sec. 5.1, Radiosonde:

- Please avoid long explanations (lines 546-574), as the well-known fact "A spatial mismatch due to radiosonde drifting would be a more plausible cause." replaces the discussion before.

  With the updated figure, this subsection has now been partly rewritten and shortened.

- There are many inter-comparisons published, but references to them are very rare (for instance to the campaign COPS; https://projekte.uni-hohenheim.de/cops/).

  True. We extended our citations by corresponding references – including COPS (l. 651-653).

Sec. 5.2, Other water vapor lidars:

- The error bars of all systems are missed.

  They are now included.

- Lines 633-635: It is only possible to present inter-comparisons which are in detail discussed including the correction term G. There is no sense for presentations of "half-inter-compared" results.

  Thank you very much for this substantial and understandable criticism.
  Purely looking at this from a theoretical standpoint, you are right that the G term should be considered. Nevertheless, there are practical reasons against the implementation of the G term.

  As is discussed at l. 192-202, the G term consists out of two terms (let's call them G1+G2). The 2nd term G2 is specially relevant in regimes where molecular backscatter dominates over particle backscatter. For lower tropospheric profiling, Bösenberg 1998 says "In most cases G2 will not have a major influence on the measurement accuracy" – due to aerosol load at those altitudes.
  For PBL measurements, the 1st term G1 is more relevant than G2 as it includes the derivation of the backscatter-coefficient-ratio, meaning that this term is big at altitudes where aerosol concentrations are changing rapidly – as it is the case for our measurements (Fig. 6b). The problem now is how to calculate this derivation which is numerically unstable for noisy signals. If one would consider a high-resolution bin-to-bin derivation, the humidity concentrations would receive an odd oscillation. If, on the other hand, signal smoothing is applied, G1 stays big in altitudes where there is actually no change in aerosol concentrations – adding an unnecessary error on healthy signals. This leads to broad "stripes" of erroneous humidity values. We made test calculations for the shown time period where we saw that the relative errors made by omitting G is around 2% in areas with no rapid aerosol change. This is below our overall statistical uncertainty and, therefore, to us, it appears to be better to neglect G rather than adding artifacts on healthy signals.
  As a second estimation, we considered Bösenbergs argumentation around Table 2 in Bösenberg 1998. There, he calculated relative errors for an assumed atmospheric aerosol load. In our specific case, the atmosphere has about a factor 5 less extinction as compared to Bösenbergs assumptions. In his Table2, this would lead

us to his "entrainment zone (EZ)" uncertainties, which for 1σ would lead to maximum relative errors of 2.7%, matching well to our test calculations.

The proper handling of G, in our opinion, calls for a special case sensitivity study on its own – which is out of scope for this manuscript.
We aren't aware of any existing work on this specific issue.

- Line 640, full saturation of the channel: I didn't get this idea. Usually, this should be avoided by the proper design of the detector coupling to the digitizer.

True. However, the ATMONSYS lidar is a non-operational, experimental system. The issue of non-linearity can already come up at 50% of the voltage range, knowledge that we gained only afterward by communication with the transient digitizer manufacturer. A respective explanation has been added (l 166-169/702-710).

- General remarks:
  I would need some more explanation regarding the topic "the paper present novel concepts, ideas, tools, or data". ·

  We have slightly rewritten the abstract and conclusion to better highlight the novel aspects of our work. Specifically, our paper presents several innovative contributions:

  - First presentation of data from new measurement tool: This is the first presentation of data from the ATMONSYS DIAL, which operates using a newly developed Titan-Sapphire (Ti:Sa) laser concept.
  - Atmospheric humidity variability: Our study addresses questions on short-term atmospheric humidity variability from the planetary boundary layer (PBL) deep into the lower free troposphere and shows:
  - High-resolution inter-comparisons: We conduct rare inter-comparisons between high-power humidity lidars with high tempo-spatial resolution.
  - New perception of Kolmogorov spectra for humidity: We introduce a new perception of "energy steps" in Kolmogorov spectra from humidity lidar, particularly in relation to heterogeneous surroundings.

  In our opinion, these topics are not yet exhausted and represent advancements in the field that are worth being published to a broad scientific community.

---

## Author Comment (AC3)

**Review comments #1:**

**Summary:**

The paper describes an instrument designed to continuously profile water vapor in the lower atmosphere at short time scales (10s averaging for Nyquist frequency of 0.05Hz). The instrument design and relevant specifications are described. An hour of high resolution data in the PBL and above are shown as examples. The goal is to resolve turbulence and an analysis of turbulence spectra are highlighted. The instrument is compared to radiosondes and state-of-the-art Raman lidar systems. Overall the paper achieves its objectives but there are some concerns and issues that need to be addressed.

Many thanks for this review. The comments helped to significantly improve this manuscript. We carefully revised the manuscript based on the comments by both reviews. When going through the individual points of criticism, we came across a small but relevant error in the previous calculation of water vapor absorption coefficients. Therefore, all plots including DIAL data were recalculated and are now in their updated version. As a result, some of the comments do not apply anymore.
A detailed explanation on the changes made, due to the recommendations and requests, is following below.

**Specific comments: major issues and concerns:**

Calibration of the DIAL needs to be discussed further. The thinking and explanation are not very scientific

- (line 87) "The DIAL technique is advantageous for measuring water vapor for several reasons, most important because it is inherently self-calibrating by its working principle". Then on line 205: The Water vapor DIAL is found to have a bias at low ranges. Then Line 210, the biases are calibrated away using radiosonde data. This removes one of the most useful benefits of DIAL, so Please discuss the magnitude of the problem, and calibration procedure further.
    - This issue is now obsolete. After the recalculation of the data, in order to meet your reasonable doubts on the correction procedure, we decided to present the data without any correction towards radiosonde profiles. Corresponding changes have been made within Sec. 2.3.
- Line 206 "However, due to the DIAL principle and the instrumental setup, this cannot be a classic overlap issue." Why not?
    - A sentence of explanation to this thought has been added to the manuscript. (l. 241-243)
- Line 207 "Therefore, we assume that there has been an issue with a detector overload which leads to this artifact." Shouldn't you be able to tell if the detector is saturating? It later is indicated that the feature that would allow the authors to tell if there are issues with clouds was turned off, is that correct?

- - An explanation to the issue of "unrecognized" detector saturation has been added at the very beginning of Sec. 2.3 and is discussed during the data analysis within Sec. 5.2.
  - The calibration makes many of the comparisons not very compelling, such as at Line 604. "4 systems show good agreement in the lowest 2 km above ground". Have not all of these systems been calibrated to the radiosonde below 1.1 km?
    - Due to the detected error in the calculation of the absorption coefficients, this valid point is now obsolete. The DIAL data isn't calibrated anymore at all. Nevertheless, you are right in the sense that the Raman lidar data has been calibrated to radiosonde ascents. However, this calibration has only been done for one radiosonde, the data is not calibrated to every new sounding.

Rayleigh-Doppler errors in DIAL

1. In the simplified version of the DIAL equation (Eq 1) it seems the outgoing and return absorption coefficients being additive is not correct.   [See Bösenberg 1998 Eq. 10 and 11]  Furthermore it is suggested to have the G term in Eq 1 written out or referenced.
   - With respect to the absorption coefficients, the given sum of outgoing and return absorption coefficients equal Eq. 11 in Bösenberg 1998. . This becomes evident within the 15[th] line after Eq. 11 in Bösenberg 1998. However, in the former manuscript version, a negative sign before the term d/dr was missing. This has now been added.
     The term G is now directly referenced to Bösenberg 1998.
1. The authors discuss that DIAL is subject to RD errors under two regimes, (molecular backscatter higher than aerosol, and at strong aerosol gradients), then proceed to not apply a correction due to its difficulty (and/or its introducing more uncertainty).  At this point, it seems relevant to note several papers in the recent literature that solve the Rayleigh Doppler problem in DIAL by simultaneously measuring the molecular to aerosol scattering ratio (backscatter ratio).  This was done with a high spectral resolution lidar (HSRL) channel.  This seems particularly applicable to the ATMONSYS instrument as it is well positioned to measure the backscatter ratio at 355 nm from the Raman lidar, or at 532 nm with an I2 HSRL channel (likely better as closer in wavelength).  Suggest that the authors acknowledge this method as a possible means to have a reliable RD correction.  This would also allow ATMONSYS to offer robustly calibrated aerosols measurements and remove reliance on Klett inversions (one of the many drawbacks of this technique shows up in Figures 6 and 7 - and noted in the caption of Figure 7)
   - Thank you very much for this interesting input. We haven't been aware of the corresponding development/publications. We've now included information on this methodology with corresponding references. (l. 200-203)
1. The above is relevant as discrepancies with the radiosondes and Raman lidars may be  due to gradients (RD error) as mentioned in line 630. And the proper implementation of the correction term calls for further research (line 634).  The "shreds of clouds" mentioned in line 635 as the most probable cause of discrepancy seems also to be gradient problems (RD errors) but this could be addressed with an HSRL channel.
   - We agree, this is something we have to consider for the future.

Line 39 Numerical Weather Prediction.  There are confusing science drivers for this instrument application.

- This instrument is well-suited for short time scales –  the science driver mentioned in the opening sentences of the abstract.  And again in line 27 understanding the humidity transport process (process studies) is a very reasonable science objective.
  - Ok.
- But the science driver given on line 41 needs clarification.  The WMO OSCAR requirements have uncertainty, temporal, vertical and horizontal requirements (note these are listed as 'goal', 'breakthrough',and 'threshold').  So is this uncertainty requirement for process studies, or for regular observations?  This instrument is not well suited to improve the numerical weather prediction for routine monitoring as it would be impractical to meet the horizontal requirements.  In the same sense that it is not economically feasible to make the radiosonde observations at sufficient horizontal spatial scales to improve weather forecasts.  Line 391 again references the WMO criteria <5%.   And finally, line 660 references this criteria again, and alludes to monitoring and data assimilation.  This does not appear to be a realistic science driver for this instrument.
  - The reference to the WMO requirements seems to have been misleading. As the ATMONSYS lidar is an experimental development, we don't claim it to be an operational instrument that immediately helps in the improvement of numerical weather prediction. However, from a developers perspective, the WMO requirements can be seen as goal system performance parameters. Recent studies incorporating single high-resolution water vapor lidars show persisting effort in the improvement of model parameters based on enhanced knowledge gained by lidar systems. Therefore, we would argue that the WMO observational requirements are a fair motivation.
    A respective sentence, clarifying that the ATMONSY lidar isn't thought to be operational, has been added (l. 42-44/736-737).

Line 197 Spectroscopic T and P dependency

- Line 197.  Please explain the rationale behind using radiosondes to inform the water vapor spectroscopic line parameters.  The data is short, presented from 1 hr of a single day.  Yet the radiosonde becomes uncorrelated in time (the radiosonde apparently was at 10.75 UTC and used to evaluate the period 11.6 to 12.6 UTC, correct?).  Would not surface measurement of the T and P (assuming a lapse rate and hydrostatic equation to get profiles) provide better results?  Reanalysis data would yield even higher quality data, if that was required.
  - T and p values over the full measurement range are important in order to precisely calculate the pressure and Doppler broadened absorption lines at different altitudes. Radiosonde data, even if it is some hours old, is a better foundation for the vertical p and T distribution than just assuming a standard lapse rate based on ground measurements. Especially in higher altitudes, one can assume that also the spatial differences are not that pronounced

anymore and persist for at least some hours (under the prerequisite of no major weather change).
We cannot answer the question on whether or not reanalysis data would be more beneficial than the up-to-date in-situ information. But sure enough, the differences should be marginal as they incorporate the radiosonde data itself.

Planetary Boundary Layer heights

- Throughout the measurement section (lines 272 - 354) the planetary boundary layer heights (PBLH) are discussed.  Moisture and aerosol gradients are used synonymously with PBLH.  But, as is well known, these methods are proxies for the PBLH and can fail for a variety of reasons.  The aerosol lidar community often overlooks this issue.  The authors have the means to measure the PBL height directly using thermodynamic buoyancy (virtual potential temperature from radiosondes) or kinematics (Doppler lidar).  For example, the vertical wind velocity in Figure 7 provides evidence that the PBL is around 1 km above ground level at 12 UTC (automated methods to derive the top of the PBLH from this data based on the bulk Richardson number exist).   At minimum, why not use these direct methods as proof that the proxy gradient methods are correct for the time shown?  This is important to provide more confidence for claims as in Line 396 "This is again an indication for the position of the PBL top"  and the analysis that follows.

    - We agree with your concerns.
    The PBL height determination purely based on the gradients of aerosol/humidity may lead to deviations in comparison to the thermodynamic/kinetic energy PBL height.
    To our understanding, a proper PBL height determination based on wind data requires horizontal wind information to identify the low level jet nose. However, the Doppler wind lidar next to the ATMONSYS system has been operated only at vertical stare. Calculating the temporal standard deviation of the vertical wind on each height leads to vertical profiles of $\sigma_w$. However, it seems to be the case that an automated PBL height determination based on this measure very much relies on personal choices for thresholds (e.g. : https://doi. org/10.16993/tellusb.1876).
    As a work around, we performed PBL height calculations based on the bulk Richardson number with the available radiosonde data. The respective values are included into the manuscript (Tab. 3).
    Above that, there was a VAD scanning Doppler lidar approx. 7 km away producing horizontal wind speed data. The sparse data that is available from those measurements, however, confirms a maximum in wind speed at around 1500m AGL during that time.
    Therefore, we trust the radiosonde data as a rough estimation of the PBL height development.
    A table with the calculated values as well as explaining sentences have been added to Sec. 3/ "Measurement day: 18 Jul 2021".

- In cases where this is not possible (perhaps the overview section around Figure 5), state that PBL heights were assumed using gradient methods and, as such, might not be the actual PBL.

- - Has been added to subsection "Measurement day: 18 Jul 2021" and all further PBL height occurrences..
- Line 370.  The rationale for the negative water vapor sounds reasonable but likely incomplete. Would not RD error be expected at the steep gradient?  How about the effect of cloud heterogeneity?  Furthermore, quality controlling the data by masking out negative water vapor might introduce problems from binning/smoothing.  Why not use a gradient method to remove clouds before retrieval of the DIAL to avoid any smoothing issues?
  - The chosen method is fairly rudimentary, but quite efficient for excluding cloud-inflicted profiles.
    Questions on biased data due to RD and/or complicated cloud heterogeneity would of course arise if the data would not be entirely neglected. However, there isn't any temporal smoothing applied beyond 10s (integration time) and the sign changing effect by signal saturation is by far stronger than any RD effects.
    The application of a cloud detection algorithm e.g. based on the computational quite intensive Haar-wavelet-transform, for this specific analysis, in our opinion, doesn't have much benefit.

Section 4.3 Turbulence spectra

- As a suggestion, since the frequency response doesn't have much overlap with the expected trend, perhaps plotting this data as an Allan Variance (two sample variance vs integration time) would be easier to interpret.  In this case the Kolmogorov constant is +⅔.
  - Thank you for this suggestion - this method hasn't been within our attention. However, for reasons of convenient inter-comparison with previous studies, we would rather stick to the chosen way of representation as the referenced publications by Wulfmeyer et al. 2024, Mauder et al. 2020 and Senff et al. 1994 use the same way of visualization.
- Of course it is possible the frequency rolloff at longer integration times may be due to instrument instability beyond 1 minute or longer.  But the most compelling rationale for the DIAL accuracy is the similar lack of low frequencies seen in Doppler winds.  The rest of the discussion regarding the reasons for the non-Kolmogorov atmosphere is tangential.
  - True. A respective sentence has been added to l. 563-566.
- Line 465 Do you mean altitude 973 m AGL and not 500 m?  It is hard to tell which altitude is deviating from which.  But it is clearly different at all altitudes  from the Doppler wind spectrum.
  - In comparison to the other lines, from our perspective, the energy drop in the 500m line seems to be most clear, but you're right that this feature can be observed in the other lines as well. The sentence has been changed accordingly.
- Line 662.  " A spectrum analysis of the DIAL showed good agreement with Kolmogorovs…"  This conclusion is unjustified.  What was shown was good agreement with the Doppler lidar frequency spectrum.  And that the Kolmogorov inertial subrange rolled off at low frequencies for some reason or other (which is not really necessary to explain)

- The respective sentence has been changed accordingly.

Technical corrections: minor grammar, misspellings, or strange word choices

Line 10. 'Evaded' perhaps 'explained' would be better?

Not sure whether "explained" is correct either, we changed it to "overcome".

Line 16.  Shreded is misspelled, but suggest changing to 'broken clouds'

Ok.

Line 46.  Deeply requested.  Suggest changing to 'often requested'

Ok.

Line 140. 'Begin of the lidar range'.  Suggest 'start of the lidar range'

Ok.

Line 166. 'Renowned DIAL equation'.  Suggest changing to 'well-known DIAL equation'

Ok.

Line 281. 'Surpassing the lidar'.  Suggest "passing over the lidar"

Ok. Has been changed at all 3 initial occurrences.

Line 336 and 633.  Straight forward should be one word, straightforward

Ok. Has been changed at all 3 initial occurrences.

Line 370 'Supersaturated'.  Not a good word choice in English for this condition.  Is it something in the electronic gain saturated or perhaps a non-linear response of the detector (perhaps some combination of both?).  Suggest 'saturated non-linear response'

Ok.

Line 379  'Spread is weaker'.  Suggest 'spread is reduced'

Ok.

Line 621. 40 min hour.  It seems the word hour is unintended

Indeed. Thanks for noticing.

Line 636 and line 675. 'Shreds of clouds'.  Suggest 'wisps of clouds'

Ok.

Line 639 'Such flags have not been set in the corresponding time',  Unclear what meaning is desired here.

> Ok, "by the transient digitizer" has been added in order to make the meaning better understandable.

Line 644 'Proof' is the wrong word. Use 'prove'

> Yes, thanks.

Line 663 'Interludes': Suggest 'portions'. Actually the spectrum doesn't agree well beyond at time scales longer than approximately 1 minute

> Ok.

**Citation**: https://doi.org/10.5194/amt-2024-168-RC1

---

## Referee Report (RR1)

Overall, the authors provided satisfactory responses in the revised manuscript. However, a few remaining issues need to be addressed. Note line numbers are from the updated (not the redlined) version of the manuscript

1.) Lines 39 - 44: The authors specifically call out the World Meteorological Organization (WMO) requirement #379 for improving High-Resolution Numerical Weather Prediction and use a subset of the 'breakthrough' levels as motivation to construct their instrument. If this science motivation is valid, why leave out the horizontal resolution requirement, in this case 5km, since observation systems have to meet all these requirements. This point is clear from the description of the scientific need (cited on the WMO website[1])

> "High-resolution (HR) Numerical Weather Prediction (NWP) focuses on observing systems required by high-resolution NWP models… The added detail is made possible by a finer computational grid on a specific area, more detailed specification of terrain, more sophisticated prescription of physical processes **mainly based on explicit rather than parameterised formulations, and, importantly, denser and more frequent observations** (with respect to global NWP) **to specify appropriately detailed initial conditions**." [emphasis added]

In the first round of reviews, I mentioned this instrument has no realistic path to meeting such a dense horizontal spacing. And the author's response was:

> "Recent studies incorporating single high-resolution water vapor lidars show persisting effort in the improvement of model parameters based on enhanced knowledge gained by lidar systems."

While this may be true, it ignores the point – development of single instruments used for parameterizations is not being requested here. This instrument does not provide the observations required to improve High-Resolution NWP and would not in the future simply due to economic considerations. So I am confused by the assertion in [line 40] that "the need for high resolution measurement data is reflected by the breakthrough requirements" of High-Resolution NWP goals. Ignoring the dense spatial observations needed to provide initial conditions is a misunderstanding of the science requirements of WMO #379.

I suggest simply dropping the references to this particular WMO goal within the following sections, as it is not relevant

Lines 39-44. "~~The need for high-resolution measurement data is reflected by the breakthrough requirements on measurement resolutions formulated by the World Meteorological Organization (WMO, https://space.oscar.wmo.int/observingrequirements) which, for "High-Resolution Numerical Weather Prediction" in the PBL are asked to be at an uncertainty level better than 5 g/kg, with vertical resolutions $\Delta z \leq 200$ m and temporal resolutions $\Delta t \leq 60$ min. Those requirements can be seen as a desired threshold for instrument development, even though the herein presented ATMONSYS DIAL is an experimental system which is not intended for routine operation~~."
* * *
[1] https://space.oscar.wmo.int/applicationareas/view/2_2_high_resolution_numerical_weather_prediction

And in the conclusion lines 738-741 "The instrumental uncertainties have been demonstrated to stay below a level of 5 g/kg, ."

2.) Lines 203 "... the ATMONSYS has been designed without an additional HSRL channel, following the thoughts of (Späth et al., 2020), effects of the Rayleigh-Doppler-broadened signal can be assumed to be quite low if the online frequency is chosen to be near the inflection point of the absorption line."
RD errors are **minimized** when at the inflection point.  But it is unclear if the instrument was operating at that point. In Figure 13 b, the DIAL data has couple sharp features which are not seen in the collocated Raman observations.  Could not these be RD errors, especially since it is said that at least one lines up with strong aerosol gradients?  This philosophy of minimizing RD errors raises a further concern.  Operating at an inflection point seems impractical since water vapor number density changes significantly from season to season.  It does not appear that ATMONSYS utilizes multiple absorption lines of varying line strength to always operate at the inflection point.  If the instrument is wavelength tuned (to optimize the optical depth for minimum relative error) based on the atmospheric conditions and seasons, Rayleigh-Doppler-broadening errors should not be ignored under all conditions. For example, if this instrument was operated in dry & cold conditions, would it be wavelength tuned away from the inflection point and reintroduce RD errors at strong gradients?   Perhaps it is better to say something like the following… For this measurement set, taken on a humid summer day when the system could be operated near the inflection point of the absorption feature, we assumed RD errors were small. But then the data shown in Figure 13b may indicate otherwise.

3.)  Line 240 "During some measurement periods, the calculated humidity profiles show an odd artifact of too high concentrations at low levels'.  And line 247, "As the reason for the behavior remains to be unclear, the presented data hasn't been modified by any correction function."
This is a significant improvement over the first manuscript, and likely more helpful to community developing DIAL instrumentation.
Line 680 "The reason for the steep humidity increase in the DIAL data towards the ground might be detector issues…"  It seems plausible that this could also be caused by stray light from the outgoing pulse.  Its decay could create a sloping baseline (e.g, during this time duration) which results in bias.
Line 706 "In contrast to Fig. 13a, the issue of unrealistically increasing humidity concentrations at the lowest altitudes is not apparent at this point of time, indicating problems with nonlinear signal behavior during daylight conditions and thin clouds."   The data shown does not fully support this statement.  While it is not as pronounced as the daytime case, it appears that the bias remains in the night case.  There is increasing humidity (~2 g/m^3) from 700m to 400 m in the DIAL data that is less pronounced (~0.75 g/m^3) in ARTHUS and radiosonde.  So the conclusion that the problem lies with the nonlinear signal behavior during daylight conditions

seems less likely.   I fully agree with what is stated in line 720 – more data is clearly needed before definitive conclusions are reached.

4.) Section 5.2.  Since the Raman lidars has been calibrated to radiosonde ascents, please state clearly in the text if the Raman lidars used the same radiosondes for calibration that is being compared in figure 13 a and b

4.) Based on the above points, I suggest a slight revising of the conclusion
Lines 758 "However, a potential problem with non-linearities caused by overload of the transient digitzer pre-amplifiers has been recognized under certain conditions as e.g. fragmented wisps of clouds passing over the lidar. This, together with steep gradients of aerosol concentration, has shown to be potentially problematic for DIAL humidity measurements "
I suggest that "caused by" could be changed to "perhaps caused in part by".  And it would be helpful to note that RD errors were not corrected in this case.

Minor grammatical errors

1. Line 4: Titanium or Ti:Sapphire not 'Titan-Sapphire'
2. Line 172:  "almost overflow from single shots"  suggest changed to "partial overflow"

---

## Author Response (AR2)

**Response to 2ⁿᵈ Review**

Overall, the authors provided satisfactory responses in the revised manuscript. However, a few remaining issues need to be addressed. Note line numbers are from the updated (not the redlined) version of the manuscript.

Thank you very much for revising our modified manuscript and your constructive suggestions on the remaining issues. We followed the reasoning of all your remarks and changed the manuscript accordingly.
A detailed response to your remarks, including a description on all changes made, is given in the following:

**1.):**
Lines 39 - 44: The authors specifically call out the World Meteorological Organization (WMO) requirement #379 for improving High-Resolution Numerical Weather Prediction and use a subset of the 'breakthrough' levels as motivation to construct their instrument. If this science motivation is valid, why leave out the horizontal resolution requirement, in this case 5km, since observation systems have to meet all these requirements. This point is clear from the description of the scientific need (cited on the WMO website)
"High-resolution (HR) Numerical Weather Prediction (NWP) focuses on observing systems required by high-resolution NWP models... The added detail is made possible by a finer computational grid on a specific area, more detailed specification of terrain, more sophisticated prescription of physical processes mainly based on explicit rather than parameterised formulations, and, importantly, denser and more frequent observations (with respect to global NWP) to specify appropriately detailed initial conditions." [emphasis added]
In the first round of reviews, I mentioned this instrument has no realistic path to meeting such a dense horizontal spacing. And the author's response was:
"Recent studies incorporating single high-resolution water vapor lidars show persisting effort in the improvement of model parameters based on enhanced knowledge gained by lidar systems."
While this may be true, it ignores the point – development of single instruments used for parameterizations is not being requested here. This instrument does not provide the observations required to improve High-Resolution NWP and would not in the future simply due to economic considerations. So I am confused by the assertion in [line 40] that "the need for high resolution measurement data is reflected by the breakthrough requirements" of High-Resolution NWP goals. Ignoring the dense spatial observations needed to provide initial conditions is a misunderstanding of the science requirements of WMO #379.
I suggest simply dropping the references to this particular WMO goal within the following sections, as it is not relevant
Lines 39-44. "~~The need for high-resolution measurement data is reflected by the breakthrough requirements on measurement resolutions formulated by the World Meteorological Organization (WMO, https://space.oscar.wmo.int/observingrequirements) which, for "High-Resolution Numerical Weather Prediction" in the PBL are asked to be at an uncertainty level better than 5 g/kg, with vertical resolutions Δz ≤ 200 m and temporal resolutions Δt ≤ 60 min. Those requirements can be seen as a desired threshold for instrument development, even though the herein presented ATMONSYS DIAL is an experimental system which is not intended for routine operation.~~"

And in the conclusion lines 738-741 "The instrumental uncertainties have been demonstrated to stay below a level of 5 g/kg,

"

> We agree with your reasoning and changed the manuscript according to your suggestions. We also adapted all subsequently occurring references to the WMO requirements within the former manuscript. Thank you.

**2.):**

Lines 203 "... the ATMONSYS has been designed without an additional HSRL channel, following the thoughts of (Späth et al., 2020), effects of the Rayleigh-Doppler-broadened signal can be assumed to be quite low if the online frequency is chosen to be near the inflection point of the absorption line."

RD errors are **minimized** when at the inflection point. But it is unclear if the instrument was operating at that point. In Figure 13 b, the DIAL data has couple sharp features which are not seen in the collocated Raman observations. Could not these be RD errors, especially since it is said that at least one lines up with strong aerosol gradients? This philosophy of minimizing RD errors raises a further concern. Operating at an inflection point seems impractical since water vapor number density changes significantly from season to season. It does not appear that ATMONSYS utilizes multiple absorption lines of varying line strength to always operate at the inflection point. If the instrument is wavelength tuned (to optimize the optical depth for minimum

relative error) based on the atmospheric conditions and seasons, Rayleigh-Doppler-broadening errors should not be ignored under all conditions. For example, if this instrument was operated in dry & cold conditions, would it be wavelength tuned away from the inflection point and reintroduce RD errors at strong gradients? Perhaps it is better to say something like the following... For this measurement set, taken on a humid summer day when the system could be operated near the inflection point of the absorption feature, we assumed RD errors were small. But then the data shown in Figure 13b may indicate otherwise.

> You're mentioning a valid point. Indeed, the wavelength has to be adapted to changing water vapor OD in order to minimize relative errors. In general, the design of our laser allows for tuning the wavelengths around the inflection point or even change to neighboring absorption lines with different strengths.
>
> For the presented data set, as you say, the wavelength has been chosen to be near the inflection point of a suitable $H_2O$ line for warm and humid conditions. In fact, the wavelength has been chosen further outside the inflection point in order to avoid that the RD-broadened light crosses over the absorption line peak – which would introduce very unwanted behavior.
>
> In order to meet your justified concerns, we reformulated the corresponding l. 198-204 for increased clarity and better understanding.
>
> The sharp features in Fig. 13B, to our opinion, are more likely due to the mentioned "wisps of clouds" (l. 707).

**3.):**
Line 240 "During some measurement periods, the calculated humidity profiles show an odd artifact of too high concentrations at low levels'. And line 247, "As the reason for the behavior remains to be unclear, the presented data hasn't been modified by any correction function."
This is a significant improvement over the first manuscript, and likely more helpful to community
developing DIAL instrumentation.

> We agree.

Line 680 "The reason for the steep humidity increase in the DIAL data towards the ground might
be detector issues..." It seems plausible that this could also be caused by stray light from the outgoing pulse. Its decay could create a sloping baseline (e.g, during this time duration) which results in bias.

> True.

Line 706 "In contrast to Fig. 13a, the issue of unrealistically increasing humidity concentrations at the lowest altitudes is not apparent at this point of time, indicating problems with nonlinear signal behavior during daylight conditions and thin clouds." The data shown does not fully support this statement. While it is not as pronounced as the daytime case, it appears that the bias remains in the night case. There is increasing humidity (~2 g/m^3) from 700m to 400 m in the DIAL data that is less pronounced (~0.75 g/m^3) in ARTHUS and radiosonde. So the conclusion that the problem lies with the nonlinear signal behavior during daylight conditions seems less likely. I fully agree with what is stated in line 720 – more data is clearly needed before definitive conclusions are reached.

> You're right that the night case is somehow ambiguous. Yes, the DIAL values are increasing at the lowest altitudes – mostly more pronounced as ARTHUS and the radiosonde. In comparison with RAMSES, however, the gradient seems to be quite the same down to 500m agl. But still, a different behavior between day and night case is obviously apparent, implying that there must be at least a partial day/night influence. To our understanding, it might well be that the problem in reality is a mixture of both daylight background and stray light from the outgoing pulse, i.e. its reflection from the outgoing mirrors.
> The respective parts of the manuscript have been slightly modified (l. 710/719-722)in order to avoid definitive conclusions which – as you say – would definitely need more data support.

**4.):**
Section 5.2. Since the Raman lidars has been calibrated to radiosonde ascents, please state clearly in the text if the Raman lidars used the same radiosondes for calibration that is being compared in figure 13 a and b

> Thank you for pointing out this missing information.
> It is now added in l. 672-677.

**5.):**
Based on the above points, I suggest a slight revising of the conclusion
Lines 758 "However, a potential problem with non-linearities caused by overload of the transient
digitzer pre-amplifiers has been recognized under certain conditions as e.g. fragmented wisps of clouds passing over the lidar. This, together with steep gradients of aerosol concentration, has shown to be potentially problematic for DIAL humidity measurements "
I suggest that "caused by" could be changed to "perhaps caused in part by". And it would be helpful to note that RD errors were not corrected in this case.

Yes. Changes within the manuscript (l. 761-765) have been made in order to meet both of your suggestions.

Minor grammatical errors
1. Line 4: Titanium or Ti:Sapphire not 'Titan-Sapphire'

Solved.

2. Line 172: "almost overflow from single shots" suggest changed to "partial overflow"

Solved.
Also, a small spelling mistake (Kolmogorovs's) in l. 745 has been corrected.